

# NOMA-MIMO in 5G network: a detailed survey on enhancing data rate

Murad Halabouni[1], Mardeni Roslee[1], Sufian Mitani[2], Osama Abuajwa[2], Anwar Osman[3], Fatimah Zaharah binti Ali[4] and Athar Waseem[5]

[1] Centre for Wireless Technology/Faculty of Engineering, Multimedia University, Cyberjaya, Cyberjaya, Malaysia
[2] Next Generation Network Research Institute, Telekom Malaysia Research & Development, Cyberjaya, Cyberjaya, Malaysia
[3] Rohde & Schwarz (M) Sdn Bhd, Cyberjaya, Cyberjaya, Malaysia
[4] Faculty of Electrical Engineering, Universiti Teknologi MARA, Shah Alam, Shah Alam, Malaysia
[5] International Islamic University, Islamabad, Pakistan

## ABSTRACT

Non-orthogonal multiple access (NOMA) is a technology that leverages user channel gains, offers higher spectral efficiency, improves user fairness, better cell-edge throughput, increased reliability, and low latency, making it a potential technology for the next generation of cellular networks. The application of NOMA in the power domain (NOMA-PD) with multiple-input multiple-output (MIMO) and other emerging technologies allows to achieve the demand for higher data rates in next-generation networks. This survey aims to funnel down NOMA MIMO resource allocation issues and different optimization problems that exist in the literature to enhance the data rate. We examine the most recent NOMA-MIMO clustering, power allocation, and joint allocation schemes and analyze various parameters used in optimization methods to design 5G systems. We finally identify a promising research problem based on the signal-to-interference-plus-noise ratio (SINR) parameter in the context of NOMA-PD with MIMO configuration.

## INTRODUCTION

### Preliminary

Future wireless networks Beyond 5G (B5G) and 6G are expected to provide high data speeds and a serve large number of users, meeting this expectation with the current access schemes is considered not achievable. High data speeds, a wide range of service needs, and ubiquitous connectivity are the goals of B5G networks. It is anticipated that there will be 10 billion user customers across the globe by 2025 (*Letaief et al., 2019*). Throughout different network generations, mobile and wireless communication networks utilize multiple access techniques, including frequency division multiple access (FDMA), time division multiple access (TDMA), code division multiple access (CDMA), and orthogonal frequency division multiple access (OFDMA). To meet future wireless network requirements, allocating resources to different users, new receiver designs, reducing interference, and expanding the number of multiplexed users accessing the spectrum (*Wei et al., 2016*;

Corresponding authors
Murad Halabouni,
murado199@hotmail.com
Mardeni Roslee,
mardeni.roslee@mmu.edu.my

**Table 1  List of acronyms.**

| | |
|---|---|
| TDMA | Time division multiple access |
| FDMA | Frequency division multiple access |
| CDMA | Code division multiple access |
| SDMA | Space division multiple access |
| OFDMA | Orthogonal frequency division multiple access |
| LoS | Line-of-sight |
| RF | Radio frequency |
| ZF | Zero-forcing |
| DoF | Degree-of-freedom |
| CSI | Channel state information |
| AWGN | Additive white Gaussian noise |
| MIMO | Multiple-input multiple-output |
| MISO | Multiple-input single-output |
| FD | Full duplex |
| OMA | Orthogonal multiple access |
| NOMA | Non-orthogonal multiple access |
| SC | Superposition coding |
| SINR | Signal-to-interference-plus-noise ratio |
| SIC | Successive interference cancelation |
| GLSIC | Group-level successive interference cancellation |
| SE | Spectral efficiency |
| BS | Base station |
| IoT | Internet of things |
| QoS | Quality of service |
| NOMA-PD | Non-orthogonal multiple access in the power domain |
| NP | Nondeterministic polynomial |
| DC | Difference of convex |
| D2D | Device-to-device |
| UAV | Uncrewed aerial vehicle |
| UL/DL | Uplink/Downlink |
| HetNet | Heterogeneous network |
| BF | Beamforming |

*Sonika, Tilak Babu & Nandan, 2021*). Table 1 shows a list of acronyms used across this work.

The current mobile wireless communication networks might not be entirely capable of maintaining user fairness and fulfilling the needs of data traffic, spectral efficiency, huge connectivity, and capacity. Orthogonal multiple access (OMA) is a leading technology that

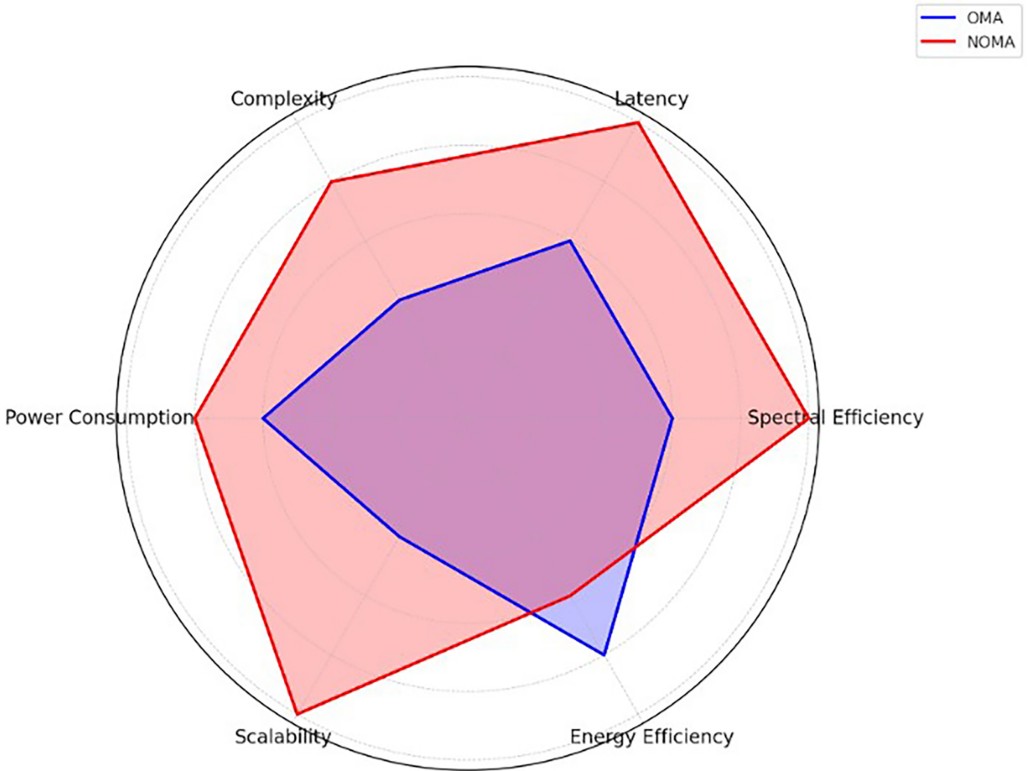

**Figure 1 Access method comparison.** Comparison of access methods based on spectral efficiency, latency, complexity, power consumption, scalability, and energy efficiency.

allows mobile communication to achieve important performance indicators such as sum rate, fairness, and cell coverage. TDMA, FDMA, and CDMA are schemes used in OMA. In OMA, the orthogonality of resources is used when users are served in the system. This orthogonality makes OMA less efficient in terms of sustaining spectral efficiency (*Saraswat & Singh, 2020*; *Dai et al., 2018*).

Before 5G's potential launch, mobile communication systems evolve to meet user demands, integrating advanced technologies once deemed futuristic. *Saraswat & Singh (2020)* introduced multiple access schemes (OMA, non-orthogonal multiple access (NOMA), delta-orthogonal multiple access (D-OMA)) for next-gen wireless systems, comparing OFDMA with NOMA's power-domain and code-domain variants. Based on *Shah et al.*'s *(2021)* work, Fig. 1 highlights the strengths and weaknesses of each access method across dimensions like spectral efficiency, latency, complexity, power consumption, scalability, and energy efficiency. OMA is moderate in most criteria, while NOMA excels in spectral efficiency, latency, and scalability but has higher complexity and power consumption.

In OMA, the base station is responsible for allocating channels and power for channels in order to serve users. In OMA, the whole available spectrum will be divided into sub-channels or sub-carriers, and each sub-channel has a fixed size and bandwidth, and a key factor is that all sub-channels must maintain orthogonality in order to avoid interference

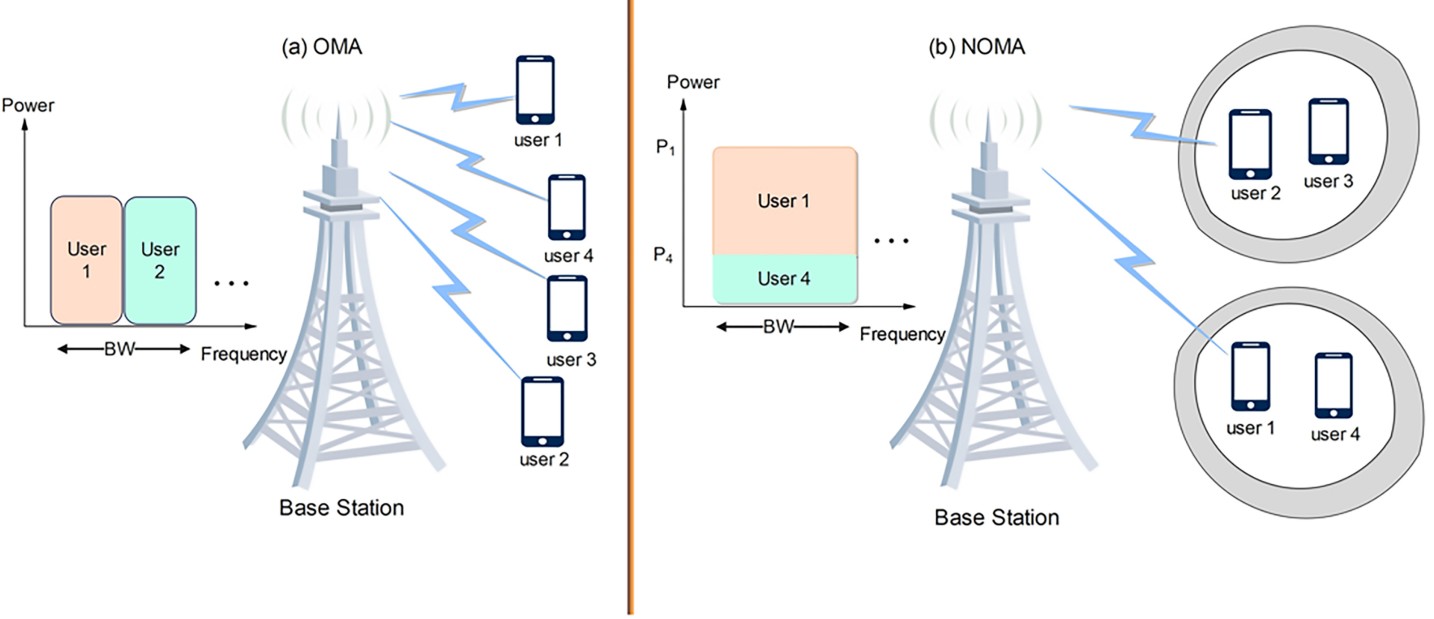

**Figure 2  NOMA OMA system.** (A) OMA system with a base station and four users scattered across the cell radius, showing different channel gains. (B) NOMA system with a base station and four users at different locations, resulting in different channel gains. Image source credit: cell tower and phone icon: Wondershare EdrawMax.               

and reduce unwanted emissions of bandwidth just to make sure that spectrum is allocated properly (*Ding et al., 2017*; *Cover & Thomas, 2001*; *Wei et al., 2017*; *Liaqat et al., 2020*; *Reddy et al., 2021*).

Figure 2A shows the OMA system, the base station with four users scattered across the cell radius which gives different channel gain, in OMA the base station will assign a channel and power level for each user and if the location of the user within the cell coverage changes then the power allocated will be the same and power level will not change. So, user 4 which is the farthest user will suffer from a degradation in service, especially in terms of data rate and delay. In OMA channels will be divided and power is fixed for each channel, and with this analogy user 4 service can be enhanced by increasing the transmitted power which is not possible in the OMA case (*Ding et al., 2017*; *Cover & Thomas, 2001*; *Jain et al., 2022*).

This has led to the appearance of NOMA, this new technique serves users in different fashions where power domain NOMA type allows for multiple users to be served using the same resource block, this allows for more efficient use of the available spectrum thus improving data rate, which leads to better performance compared to conventional OMA schemes. This is where NOMA in the power domain (NOMA-PD) comes to tune the power ratio in order to provide a better service for users who are located at the edge or users suffering from bad service. Also, Fig. 2B shows the NOMA system, in NOMA the base station has four users in its coverage area, each user is located at a different location and thus has a different channel gain (*Cui et al., 2022*).

## Scope of this survey

The goal of our survey is to not only review solutions for resource allocation, concepts, and hot emerging technologies related to NOMA-PD, but the key motive behind this work is to provide extensive work done on enhancing data rate using NOMA-multiple-input multiple-output (MIMO). This work discusses and analyzes recent resource allocation techniques and tools used to optimize and improve performance by elaborating on clustering, power, and interplay with new emerging technologies such as beamforming, device-to-device (D2D), and heterogenous network (HetNet).

## Survey contributing

The survey extensively covers NOMA-PD and NOMA-MIMO solutions, explores progress in resource allocation techniques, focuses on key questions related to MIMO applications in NOMA, highlights the importance of signal-to-interference-plus-noise ratio (SINR), and underscores NOMA's critical role in meeting the demands of 5G technology.

- Comprehensive review of NOMA-PD and NOMA-MIMO solutions, focusing on resource allocation, emerging technologies, and optimization techniques.
- Exploration of recent advancements in resource allocation methods, including clustering, power allocation, and integration with emerging technologies like beamforming and D2D.
- Framing survey methodology around key questions regarding MIMO applications in NOMA, challenges in data rate improvement, and effective schemes for MIMO-NOMA.
- Emphasis on the significance of SINR in NOMA, highlighting its role in user pairing, clustering, and power allocation.
- Identification of NOMA as a pivotal technology for addressing 5G demands, with integration with MIMO to enhance coverage, reduce interference, and maximize network capacity.

## Survey methodology

The survey scope and methodology are based on key questions regarding MIMO applications in NOMA, challenges in data rate improvement, effective schemes for MIMO-NOMA, and the impact of SINR, channel gain, and distance on MIMO-NOMA performance. The following questions formed our survey scope and method and we aimed to tackle them throughout this survey:

- What are the MIMO applications in NOMA?
- What are the challenges to improving the data rate in NOMA-MIMO?
- What is the most effective scheme to be used in MIMO-NOMA?
- How does SINR, channel gain, and distance affect the performance of MIMO-NOMA?
- What work and schemes have been proposed to solve issues in MIMO-NOMA?

## Audience of this survey

This survey targets researchers, engineers, and professionals, focusing on the advancement and optimization of NOMA technologies within wireless communication systems. It provides a detailed examination of NOMA concepts, advantages, and challenges, particularly in the context of integrating NOMA with emerging technologies like multiple in MIMO, D2D, beamforming, and HetNets. The audience is assumed to possess a technical background in wireless communication systems and optimization techniques, as the text discusses resource allocation algorithms, clustering strategies, power control methods, and SINR optimization. It offers insights into recent research findings and identifies open research problems, aiming to guide future work in enhancing data rates and system performance in 5G NOMA-MIMO.

## Data sources and research strategy

To complete our survey several common data sources were used in research on NOMA and its applications in wireless communication systems. These sources include academic research articles and journals covering NOMA, MIMO systems, and 5G networks, providing insights into optimization techniques and resource allocation algorithms. Examples of journals used such as IEEE Transactions on Wireless Communications, IEEE Xplore, Google Scholar, and arXiv.

The inclusion/exclusion criteria employed in the text involve conducting a comprehensive survey to explore the application of NOMA technology, particularly in conjunction with MIMO systems and other emerging technologies, to enhance data rates in next-generation wireless networks. The research criteria are primarily focused on selecting literature and studies that address NOMA-MIMO resource allocation issues, optimization methods, and challenges related to improving data rates.

## Organization of the paper

This survey is structured as follows. "NOMA Concept" discusses NOMA system design, then introduces the NOMA concept along with key working technologies, and ends with power domain NOMA. "NOMA-MIMO" discusses NOMA-MIMO and key performance indicators along with other emerging technologies considered crucial in modern wireless networks and ends with a brief of the algorithm used. "SINR" introduces the research gap and open research problem in NOMA-MIMO Data-rate enhancement. Lastly, "Conclusion" concludes the article.

## NOMA CONCEPT

NOMA-PD uses the available spectrum in an efficient way that allows for the spectrum to be used without the need to divide it, thus orthogonality and the waste of available spectrum that comes with it is avoided, also the ability to change the power ratio between two users grouped together enhances the service for edge cell users which translate to a better data rate and lower delay and lower outage probability (*Islam et al., 2016*; *Huang et al., 2019*; *Zhu et al., 2017*). Various power allocation algorithms are designed to improve system efficiency, total rate, and user fairness (*Ganti et al., 2010*; *Le et al., 2019*). Most of

the literature on NOMA is based on NOMA-PD, which implies a strong power distinction between the signals allocated to various NOMA users. All users' signals are allocated by superposition coding (SC). In a NOMA system which is populated by the number of users, the order of the received signal is based on normalized channel gain of each user signal (*Khansa et al., 2019*; *Lin, Tang & Ghassemlooy, 2019*). In *Khan et al. (2020)* and *Huang et al. (2018)* two NOMA users on a single channel will experience perfect fairness when the optimal power allocation satisfies the max-min fairness requirement. In order to obtain higher successive interference calculation (SIC) performance a gap between users' signal strength has to be maintained (*Islam et al., 2016*; *Fang et al., 2017b*; *Wei, Ng & Yuan, 2016*).

## NOMA system

Efficient resource allocation within NOMA is achieved by allowing users to be allocated in the same sub-channel. In the NOMA system base station (BS) is responsible for assigning users to sub-channels and also for allocating power between users who share the same sub-channel (*Huang, Wang & Zhu, 2018*; *Fang et al., 2017a*; *Sun et al., 2017*). Figure 2B depicts a downlink NOMA network the BS transmits a signal to M user device through N sub-channel, and SIC is used at the receiver side. In NOMA-PD the base station will assign two users on the same channel, which is translated to two users allocated the same frequency at the same time. User 1 and user 4 will be allocated to channel 1 and user 2 and user 3 will be allocated to channel 2. The power for channel 1 will be $P_n = P_{1,1} + P_{1,4}$ and the power for channel 2 will be $P_n = P_{2,2} + P_{2,3}$. And what NOMA does is always try to balance the power ratio between user 1 and 4, to ensure fair power distribution, and user 4 whose location is the farthest receives the needed power to provide the needed service. The correct tuning and manipulation of power between user 1 and 4 will result in a noticeable enhancement for user 4 and 1 (*Reddy et al., 2021*).

Figure 3 shows the NOMA-MIMO concept and user clustering along with power allocation tuning. The user device is represented by m, where $m \in (1, 2, 3, , M.)$ and the sub-channel $(SC_n)$ is represented by n where $n \in (1, 2, 3, , N.)$. The total available bandwidth is divided between sub-channels where each sub-channels bandwidth can be expressed as follows $B_{SC} = \dfrac{BW}{N}$ (*Vanka et al., 2012*). In NOMA users are allocated across all sub-channels where we will denote them in $(M_N)$, and the power allocated to any user across any sub-channel will be denoted in $(P_{I,n})$, and we can write power allocated on a given sub-channel below.

$$\sum_{i=1}^{Mn} P_{i,n} = P_n. \tag{1}$$

The total transmitted power from the base station ($P_s$) can indicate the constraints of power for all sub-channels ($P_n$), and then the total transmitted power from the base station can be expressed in Eq. (2) (*Jain et al., 2022*).

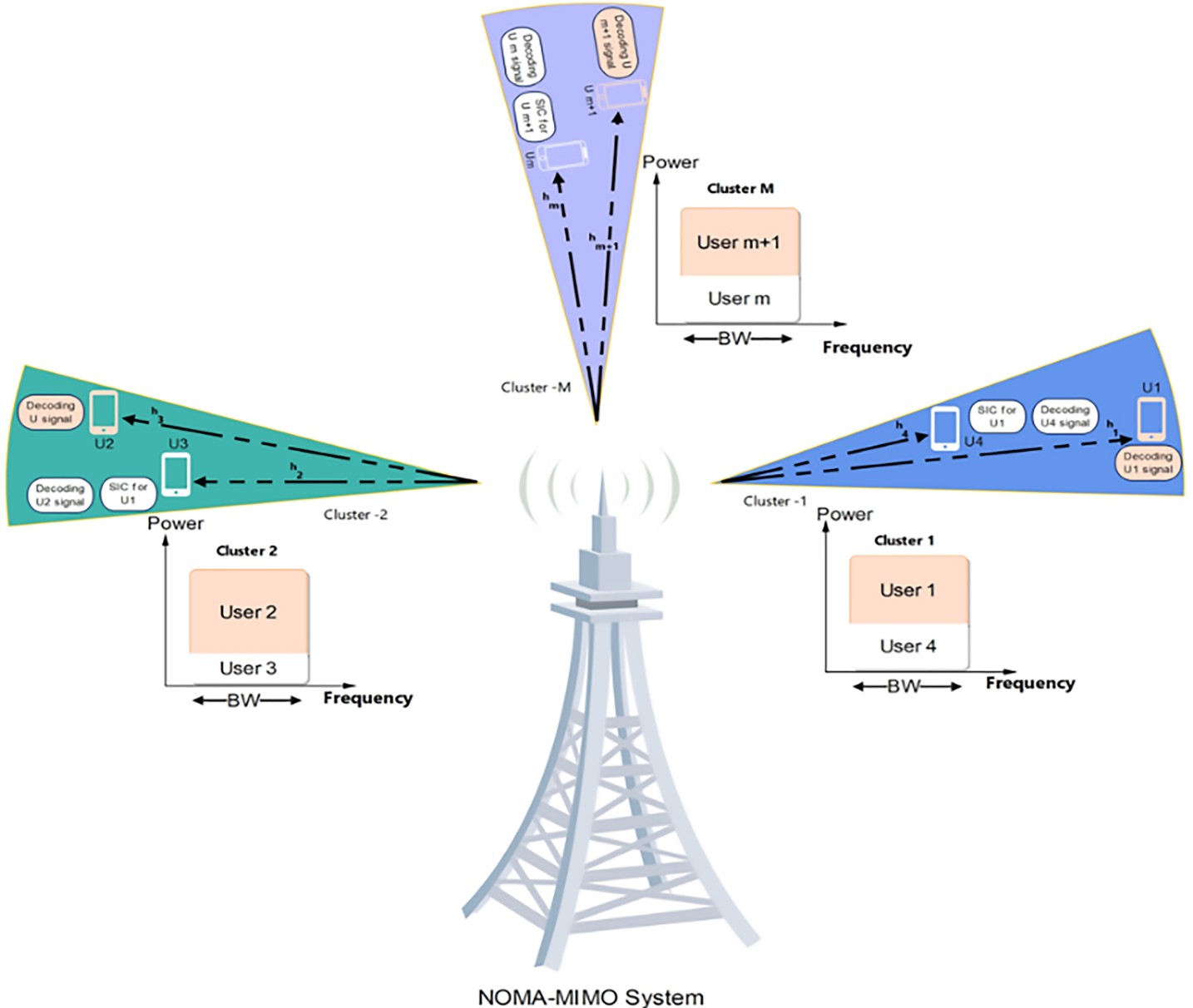

**Figure 3 Noma-MIMO system.** NOMA-MIMO concept showing user clustering and power allocation tuning, with user devices represented by 'm'. Image source credit: cell tower and phone icon: Wondershare EdrawMax.

$$\sum_{n=1}^{N} P_n = P_s. \tag{2}$$

In power domain NOMA the base station will allocate users with different power levels across all sub-channels. We can write the transmitted signal as follows.

$$x_n = \sum_{i=1}^{Mn} \sqrt{P_{i,n}}\, S_i \tag{3}$$

where $S_i$ is the data symbol of the user ($i$) on $SC_n$, and $z_{l,n}$ is noise affecting the received signal. The received signal can be expressed as follows.

$$y_{l,n} = h_{l,n}\, x_n + z_{l,n}. \tag{4}$$

$$= \sqrt{P_{l,n}}\, h_{l,n}\, S_i + \sum_{i=1\ i\neq l}^{Mn} \sqrt{P_{l,n}}\, S_i + z_{l,n}. \tag{5}$$

The term $\sqrt{P_{l,n}}\, h_{l,n}\, S_i$ represents the user message, and $\sum_{i=1\ i\neq l}^{Mn} \sqrt{P_{l,n}}\, S_i$ is other users' messages on the same sub-channel where $i \neq l$ is all users except user $l$. Where $h_{l,n}$ is the coefficient of the sub-channel number ($SC_n$) from the base station to the user device, and a channel gain fading is assumed within $(h_{l,n})$. In NOMA systems SIC is used at the user side to extract the desired signal. For user l using SIC, we can write the SINR as follows (*Jain et al., 2022*).

$$SINR_{l,n} = \frac{P_{l,n}\, H_{l,n}}{1 + \sum_{i=1}^{l-1} P_{l,n}\, H_{l,n}}. \tag{6}$$

According to the Shannon rule, we can write data rate for user l on sub-channel n as follows.

$$R_{l,n} = [P_{l,n}] B_{SC} \log_2 \left[ 1 + \frac{P_{l,n}\, H_{l,n}}{1 + \sum_{i=1}^{l-1} P_{l,n}\, H_{l,n}} \right]. \tag{7}$$

And the overall sum-rate of the system (*Jain et al., 2022*) can be written as,

$$R = \sum_{n=1}^{N} \sum_{l=1}^{Mn} R_{l,n}\left(P_{l,n}\right) = \sum_{n=1}^{N} R_n\left(P_n\right) \tag{8}$$

when spectrum availability is compared between NOMA and OMA, the literature shows that NOMA has higher system throughput, while in OMA the available spectrum is divided between users, and each user is allocated a single frequency. In NOMA the spectrum is available for each user for transmission which results in higher system throughput (*Liaqat et al., 2020*; *Reddy et al., 2021*; *Cui et al., 2022*). *Al-Abbasi & So (2016)* investigate maximizing the total data rate in a NOMA system that operates on frequency-selective fading channels. Users are grouped according to their channel strengths, and a hierarchical method for power allocation is introduced. This grouping allows for a straightforward calculation of power allocation. The procedure continues until all users have been assigned their transmission power. This framework is designed to accommodate a large number of users in NOMA systems.

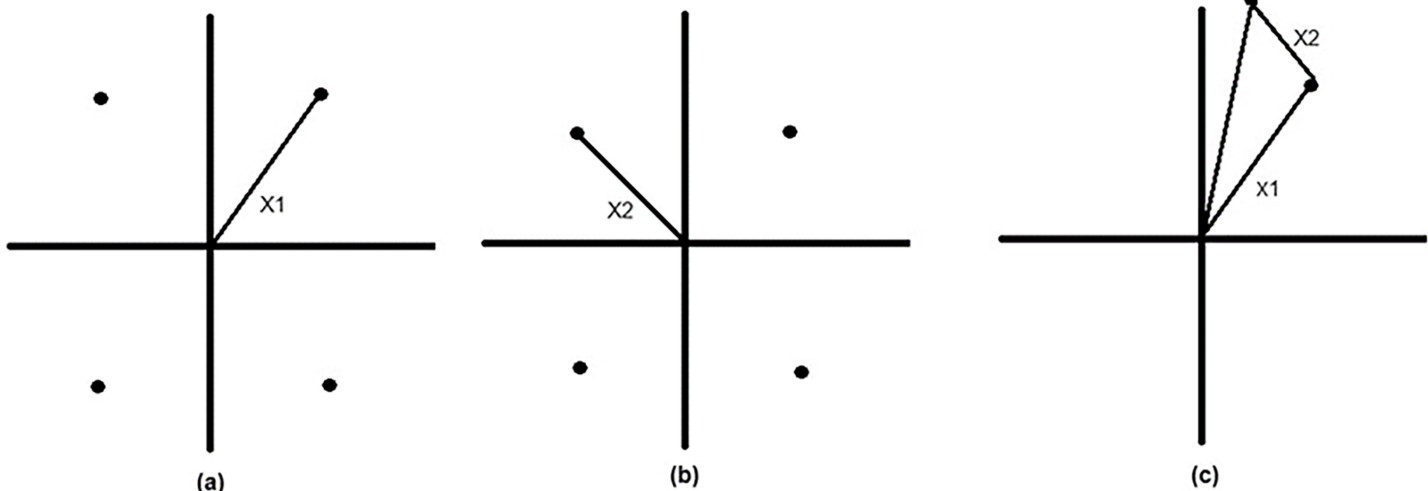

**Figure 4** **Superimposed signal.** (A and B) Quadrature phase-shift keying (QPSK) constellation for two users. (C) Constellation of superimposed signal.

## NOMA concept

The two primary methods of NOMA are SIC and SC. NOMA's key benefit is its ability to serve multiple users simultaneously from the base station by utilizing superposition coding, in which a power coefficient is assigned to each user and the transmitter superimposes the signals of the users (*Vanka et al., 2012*; *Otao, Kishiyama & Higuchi, 2012*; *Miridakis & Vergados, 2013*; *Vizi et al., 2011*).

For a two-user transmission, the transmitter will map the users signals as a complex-valued sequence to an encoder. Figures 4A and 4B show the quadrature phase-shift keying (QPSK) constellation of the two users where user 1 has higher power than user 2, due to the fact that user 1 is considered farther than the user, Fig. 4C shows constellation of superimposed signal. *Huang et al. (2018)* proposed a superposition encoding technique by using off-the-shelf single-user coding and decoding blocks $S_1(n)$.

In the SC phase, two point-to-point encoders representing the two users' signals $X_1$ and $X_2$ are mapped to two output bit sequences $S_1(n)$ and $S_2(n)$, respectively. $f_1:\{0,1\}^{[2TR1]} \rightarrow C^T$ is mapped to $S_1(n)$, and $f_2:\{0,1\}^{[2TR2]} \rightarrow C^T$ is mapped to $S_2(n)$, R1 and R2 represent the transmission rates of both users and each has a block of length T, and C is a code library; then to complete the superposition encoding (*Cui et al., 2022*) a summation device gives output as follows,

$$X_n = \sqrt{P\,\beta_1\,S_1(n)} + \sqrt{P\,\beta_2\,S_2(n)}. \tag{9}$$

The $\beta_x$ represents a portion of the total power P assigned to the $user_x$ while keeping the constraint of $\beta_1 + \beta_2 = 1$. All signals are ordered at the receiver in accordance with the strength of the signal that was received. The strongest signal is first decoded and subtracted from the incoming superimposed signal. In order to extract the desired signal, the user

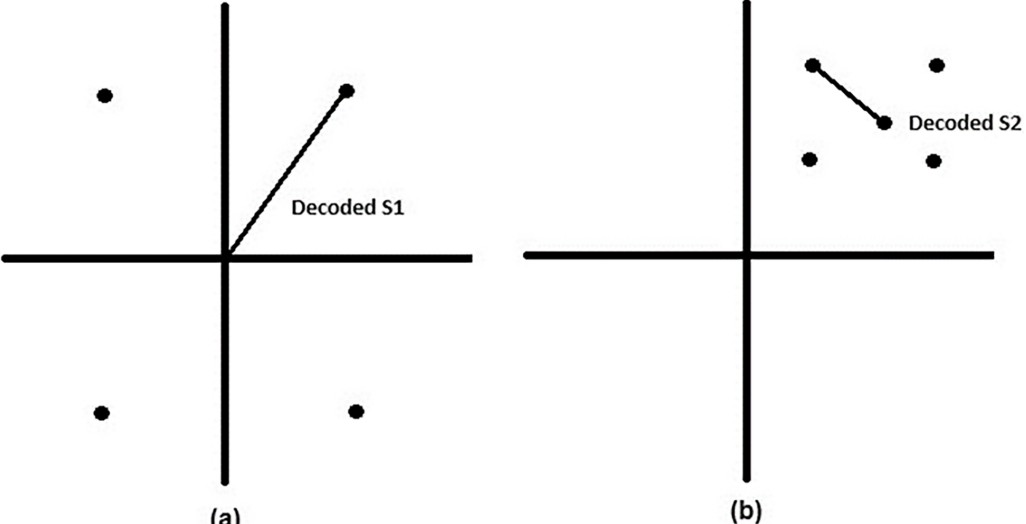

**Figure 5 Decoding of duperimposed signal.** (A) Received signal of user 1 with its decoded constellation point. (B) Decoded constellation point of user 2 relative to user 1's decoded point.

must perform this action repeatedly, this process is called SIC (*Huang, Wang & Zhu, 2018*; *Fang et al., 2017a*; *Sun et al., 2017*; *Abuajwa et al., 2022*; *Abuajwa, Roslee & Yusoff, 2021*).

The decoding of the superposed signal is shown in Fig. 5. Figure 5A represents the received signal of user 1 constellation point is decoded, then in Fig. 5B the constellation point of user 2 is decoded in relation to the decoded constellation point of user 1. In conjunction with NOMA one of the main problems with NOMA systems is computing and processing complexity, which can be reduced by using SIC on the receiver side. The accompanying coding and decoding error lowers the system's performance. Users with poor channel conditions in an OMA system, typically receive poor-quality channel and experience service disruptions.

The NOMA system's power allocation improves data rate, fairness, and quality of service (QoS) by utilizing strong channel conditions as relays to increase data rate for poorer channels. The order of the users in NOMA is determined by the channel strength to help users cancel others' interference, fairness is achieved by ranking users and giving user with poor channel conditions more power (*Ali, Tabassum & Hossain, 2016*; *Zamani, Eslami & Khorramizadeh, 2018*). Also, *Abuajwa et al. (2022)* evaluated the rate difference among users in order to achieve data fairness, the scheme for a fairness system based on the non-uniform power distribution. The user's data rate is calculated using a portion of the total power allocated to them, and this is done while continuously monitoring the fairness index, which is set at 1, meaning that fair rates are reached. *Islam et al. (2016)*, *Zhu et al. (2017)*, *Le et al. (2019)*, and *Huang, Wang & Zhu (2018)* introduced the direct effect between the number of users that can be served and cell radius, where the distance between users and the base station can affect channel gain and thus path loss, so the number of admitted users is related to cell radius, and number of users per cell is an important factor to control throughput balance to achieve fairness.

## NOMA power domain

The power allocation assignments utilized in NOMA maintain user equality (*Rezvani et al., 2021*). Users are ordered according to their channel condition; power is distributed accordingly in order to ensure a proper channel gain (*Glei & Chibani, 2020*). NOMA-PD is considered an optimal solution due to the efficient use of available resources in contrast to OMA where the available orthogonal resources are limited, NOMA-PD enhances connectivity by assigning a single frequency channel to serve multiple users inside one cell. Thus, offering massive connections when compared to OMA (*Lei et al., 2016*; *Ding et al., 2017*; *Lei et al., 2016*; *Salauen, Chen & Coupechoux, 2018*). In traditional OMA systems, access-grant requests are used to admit users into the system, and the BS responds to scheduling requests from users by sending the clear-to-send signal. In addition, a higher system throughput is provided by NOMA-PD, since NOMA-PD does not divide available bandwidth among users allocated together, and each user can use all of the available spectrum (*Kim, Jafarkhani & Lee, 2022*). In addition, in NOMA-PD scheduling requests from users toward the BS is not required because NOMA-PD serves users simultaneously. Thus, in NOMA-PD scheduling is not required, which results in a lower transmission latency (*Huang et al., 2019*; *Gupta & Ghosh, 2020b*; *Norouzi, Champagne & Cai, 2023*; *Thet & Ozdemir, 2020*; *Salaun, Coupechoux & Chen, 2020*; *Haq & Taspinar, 2021*; *Hanif & Ding, 2019*; *Gupta & Ghosh, 2023*; *Gupta & Ghosh, 2020a*; *Hong et al., 2020*). In OMA, each user is allocated a dedicated channel and channel power which is part of the orthogonal resource allocation. While OMA can serve a single user at a time, the NOMA scheme can serve multiple users at once. As a result, in terms of spectral efficiency, system throughput, data rate, and quality of service, the NOMA system is superior to the OMA design.

*Gupta & Ghosh (2023)* introduced a mixed-integer non-linear programming (MINLP) nondeterministic polynomial (NP)-hard optimization problem for mobile users, focusing on equal power distribution in a two-user NOMA system with sequential resource allocation. Building on this, *Gupta & Ghosh (2020b)* investigated maximizing transmitted power in multi-carrier NOMA (MC-NOMA) systems using joint sub-channel and power allocation methods, addressing energy ratio concerns. *Huang et al. (2019)* presented two schemas to enhance NOMA system throughput by optimizing power allocation and system sum rate through convex programming algorithms and perfect channel state assumptions.

*Gupta & Ghosh (2023)* proposed a channel assignment strategy and optimal power allocation scheme under constraints of power budget and QoS and the mobile user is allocated in one channel. The nature of the problem was a MINLP NP hard optimization problem. To achieve maximization of NOMA system throughput and sum rate, authors proposed equal distribution of initial power to all users and a two-user NOMA system where a joint problem of channel selection and power allocation is proposed, and authors divided the problem into two sequential resource allocation approaches. For channel assignment of users, users are accommodated on a channel based on their channel condition in order to maximize system throughput, due to the fact that SINR is directly proportional to the data rate allocation problem is considered a binary linear

programming problem. As for the power allocated of users assigned on a channel, authors solved the non-linear problem of a proposed objective function that is concave in nature due to channel assignment strategy which depends on channel condition and thus SINR, interior point method (IPM) is used to obtain power allocation coefficients. for each channel, a power allocation coefficient vector is produced using IPM by dividing the allocation problem into sequences and iteratively solving the problem.

*Gupta & Ghosh (2020b)* tackled maximizing the transmitted power through joint sub-channel and power allocation for the MC-NOMA systems. The authors' schema allocates users into the sub-channel by the matching algorithm. Next, their maximization problem which is divided into sub-problems is solved by the penalty function method. The Dinkelbach algorithm is proposed as the joint channel and power allocation algorithm. For the matching algorithm, users ordered based on channel conditions and then assigned to sub-channels, authors calculated the minimum power for all users under the minimum data rate required this was important to make sure all users to be allocated did not exceed the maximum power from the base station, the minimum rate constraint is considered a concave function which made power allocation problem a convex optimization problem. The maximization problem the authors proposed is a mixed non-convex problem with maximum transmit power and minimum rate requirement of each user constraint since energy is considered as a ratio of the total achievable rate to the total power consumption of the system. *Huang et al. (2019)* proposed two schemas in the efforts to improve NOMA system throughput, due to the relationship between power allocation and system sum rate authors investigated two methods for power assignments to users, the first is distributing power to all sub-channels equally and the second is using the difference of convex (DC) programming algorithm for power allocation. Due to the inversely proportional relationship between power and channel gain their work constraints are to ensure a minimum data rate and to guarantee the maximum power transmitted from the base station. The authors divided the problem into two parts, firstly the allocation of users into sub-channels and the second is power allocation among sub-channels. As for the allocation of users, authors considered a matching schema between sub-carriers and users by creating a matrix and considering a perfect channel state, the rows of the matrix represent sub-carriers and the column represents users and the order is with respect to the channel gain of the users. Next comes the power allocation, the authors' first schema is Equal power allocation for all users across sub-channels, and in this way, the sum-rate is depending on the user's channel gain. In the second schema where the DC programming algorithm is implemented, a power factor value between (0, 1) is searched for and weighted bandwidths are multiplexed by this factor while considering users' channel gain. The result of their work with the DC programming algorithm provided a better sum-rate. The work (*Gupta & Ghosh, 2020b*; *Norouzi, Champagne & Cai, 2023*; *Thet & Ozdemir, 2020*; *Salaun, Coupechoux & Chen, 2020*; *Haq & Taspinar, 2021*; *Hanif & Ding, 2019*; *Gupta & Ghosh, 2023*; *Gupta & Ghosh, 2020a*; *Hong et al., 2020*) is extensively elaborated to demonstrate the process of achieving objectives through handling resource allocation problems with different algorithms. Table 2 shows a summary of work done on power allocation.

**Table 2 Summary of works on power allocation.**

| Ref. | Objective | Algorithm and type of the problem | | Constraints | Outcomes |
|---|---|---|---|---|---|
| *Lei et al. (2016)* | Channel assignment strategy, optimal power allocation scheme | MINLP NP hard | Mixed-integer Non-convex | Power budget. QoS | Maximization of throughput and sum rate |
| *Salauen, Chen & Coupechoux (2018)* | Maximizing power allocation and energy efficiency | Matching algorithm. Penalty function. | Mixed non-convex problem | Maximum power transmitted. Minimum data rate required | Effective resource allocation algorithm |
| *Kim, Jafarkhani & Lee (2022)* | Improving NOMA system throughput | DC programming algorithm | Non-convex | Minimum data rate. Maximum power transmitted. | Better sum-rate by DC programming |
| *Gupta & Ghosh (2020b)* | Optimize resource allocation in DL NOMA system | Dinklebach algorithm. | Concave objective function NP-hard | Maximize EE. Minimum QoS | Maximize overall energy efficiency |
| *Norouzi, Champagne & Cai (2023)* | Maximizing sum-rate for OR and PA in a partial NOMA system | Gradient search algorithm | Search algorithm | User fairness | Higher sum rate. Satisfying Jain's fairness levels |
| *Salaun, Coupechoux & Chen (2020)* | Less complexity of user selection and transmit power | i-SCPC i-SCUS | NP-hard | Power constrain | Sum-rate maximization |
| *Haq & Taspinar (2021)* | Power allocation to enhance SINR balancing | GSVD along with three uncertainty models | Convex formulations | Transmit power | Robust performance |
| *Hanif & Ding (2019)* | Enhance the spectral efficiency | Quadrature spatial modulation | Jointly resource allocation problem | Power constrain | Enhance bit error rate (BER). Better sum rate |

**Note:**
MINLP, mixed-integer non-linear programming; NP, nondeterministic polynomial; QoS, quality of service; NOMA, non-orthogonal multiple access; DL NOMA, downlink NOMA; EE, energy efficiency; OR, overlap ratios; PA, power allocation; i-SCPC, single channel per carrier; i-SCUS, single-carrier user selection; SINR, signal-to-interference-plus-noise ratio; GSVD, singular-value decomposition; BER, bit error rate.

In a similar vein, *Haq & Taspinar (2021)* utilized an iterative approach to optimize power allocation in downlink NOMA cellular systems, aiming to maximize system throughput and data rate. *Gupta & Ghosh (2020a)* proposed a joint allocation algorithm for overlap ratios and power allocation in partial NOMA systems, aiming to balance sum-rate and user fairness. *Thet & Ozdemir (2020)* tackled the joint optimization of user clustering, beamforming, and power allocation schemes using a MINLP approach and branch-and-bound algorithms. *Norouzi, Champagne & Cai (2023)* introduced new algorithms to reduce the computational complexity in user selection and single-carrier power control methods, aiming to maximize the weighted sum-rate while minimizing complexity. *Salaun, Coupechoux & Chen (2020)* integrated singular value decomposition (GSVD) in NOMA-MIMO systems to handle channel estimation errors and channel uncertainty, employing different models to balance SINR. Finally, *Hanif & Ding (2019)* proposes a quadrature spatial modulation (QSM) in NOMA-MIMO systems to improve spectral efficiency and introduces a low-complexity user grouping and power allocation schema. By a power allocation strategy, *Haq & Taspinar (2021)* managed to maximize system throughput while meeting the minimum required data rate within the base station's limited power budget. An iterative approach is proposed to solve this problem, authors

used the Dinklebach method to optimize the resource allocation scheme for a downlink NOMA cellular system which led to reducing the objective function's complexity and converting the non-linear fractional programming problem into a linear problem. Authors in their model used distance measurement to represent the received signal from the BS. The authors formed a joint optimization problem to solve the channel assignment problem and optimize power allocation coefficients for each channel. For channel assignment, the authors used a cvx 2.0 solver and a two-sided matching algorithm to solve assigning channels to users to maximize system throughput, and the power allocation coefficients are obtained by using the IPM. *Norouzi, Champagne & Cai (2023)* proposed a joint allocation algorithm for overlap ratios (OR) and power allocation in a partial NOMA system (P-NOMA) with the objective of maximizing the sum-rate. The constraint the authors used is user fairness. The proposed algorithm utilizes bandwidth efficiently by allowing the base station to transmit both OMA and NOMA signals through overlapped regions, the proposed system has two users, and joint allocation of overlapped regions and power allocation coefficient are defined under Jain's fairness index, for each user a fraction of bandwidth is dedicated through the pre-mentioned variables. under the objective of maximizing the sum rate and achieving user fairness, the authors formulated an optimization problem, a gradient search algorithm used to lessen the complexity of the system. *Thet & Ozdemir (2020)* proposed a MINLP that is used to formulate the joint optimization framework for user clustering, downlink beamforming, and power allocation schemes. The goal of MINLP is to minimize the total transmission power while meeting QoS, user clustering, and power constraints. The investigated problem is highly complex due to being non-convex and combinatorial. To address high complexity, authors developed a branch-and-bound (BB) algorithm that controls the possible steps in a given space and constrains the objective function, and even with the BB algorithm proposed by the authors, they managed to obtain an ε-optimal solution within a limited number of iterations. Consequently, they reformulated the problem to make it more tractable and devised a simplified algorithm utilizing the penalty dual-decomposition technique. In their study, the decoding order of users is determined by their effective channel gains. Moreover, they ensure successful SIC by guaranteeing that the SINR of a user's decoded signal is greater than or equal to that of another user's signal within the same group. *Norouzi, Champagne & Cai (2023)* reduced the computational complexity of current user selection and single-carrier power control methods with new algorithms that were created to compute an optimal solution with less complexity. Authors suggested ε-JSPA, a fully polynomial-time approximation scheme (FPTAS) to minimize the complexity and maximize the weighted sum-rate (WSR). The authors' work introduces two algorithms, the single channel per carrier (i-SCPC) and single-carrier user selection (i-SCUS), to solve the Pseudo-Polynomial Time Optimal Scheme (JSPA) problem which achieves a controllable and tight trade-off between approximation of single-carrier transmit power control guarantee and complexity. *Salaun, Coupechoux & Chen (2020)* introduced GSVD in the NOMA-MIMO system to tackle channel estimation errors and integrate channel uncertainty in power allocation problems to obtain SINR balancing. The authors addressed power allocation to balance SINR in a two-user system scenario. Authors considered three

different types of models taken into account in the power allocation strategy for MIMO-NOMA systems. These models incorporate channel uncertainty into the variances of the error matrices and enable tractable formulations of the SINR balancing problem. The authors introduce the Deterministic Uncertainty Model, which assumes deterministic channel estimation errors with fixed error matrices, formulating the power allocation problem based on these fixed error values. To form power allocation constraints authors used the ellipsoidal uncertainty model which concentrates on channel estimate errors that are bounded inside an ellipsoidal region. To make the problem convex and to achieve SINR balancing authors used the polyhedral uncertainty model to create a convex region in which the constraints matrices can vary. *Hanif & Ding (2019)* proposed a QSM in the NOMA-MIMO system in order to enhance the spectral efficiency. Also, a low-complexity user grouping and power allocation dynamic schema was proposed. In their work authors used channel conditions, user locations, and channel correlations to perform user grouping, and power allocation is decided based on user rate requirements and QoS. For clustering authors depended on the chordal distance between two users. The chordal distance reduces with the distance of users to one another then the average chordal distance is calculated for all users in the same cluster. As for the power allocation scheme, based on the rate requirement of each user power was adjusted to make sure each user could correctly detect the symbols modulated while also satisfying the target-rate requirement.

The research area in NOMA-PD systems is rich and diverse, spanning various optimization techniques and algorithms to enhance system performance. From MINLP formulations to iterative approaches and novel algorithms, researchers have tackled complex challenges such as power allocation, user clustering, beamforming, and channel estimation errors. These efforts aim to maximize system throughput, sum-rate, user fairness, and spectral efficiency while minimizing computational complexity. Overall, the collective efforts presented in the cited works demonstrate the continual evolution and advancement of the NOMA-PD system, promising improved performance and efficiency in future wireless communications.

## NOMA-MIMO

The work introduced in "NOMA Concept" on NOMA focused on a single antenna, when MIMO was introduced and used in NOMA an increase in performance and improvement was noted (*Rajaraman, 2002*; *Hatoum et al., 2011*; *Joud, Garcia-Lozano & Ruiz, 2018*; *Cui, Ding & Fan, 2018*; *Kumar et al., 2019*; *Gimenez, Michaelsen & Pedersen, 2016*; *Rihan, Huang & Zhang, 2018*; *Asadi, Wang & Mancuso, 2014*; *Zeng et al., 2017a*; *Liu et al., 2018*; *Ali, Hossain & Kim, 2017*; *Zhu, Zhao & Zhou, 2018*), and the improved gain resulted from MIMO allowed for more degree-of-freedom (DoF). The correlation of NOMA and MIMO allowed for efficient utilization of the available spectrum based on the spatial multiplexing order of MIMO. *Huang et al. (2018)*, *Zeng et al. (2017b)*, *Salaün, Coupechoux & Chen (2019)*, and *Rezvani & Jorswieck (2022)* concluded that MIMO-NOMA showed better results in terms of sum channel capacity and ergodic capacity when compared to MIMO-OMA while considering several users in one cluster, authors concluded that the sum-rate is

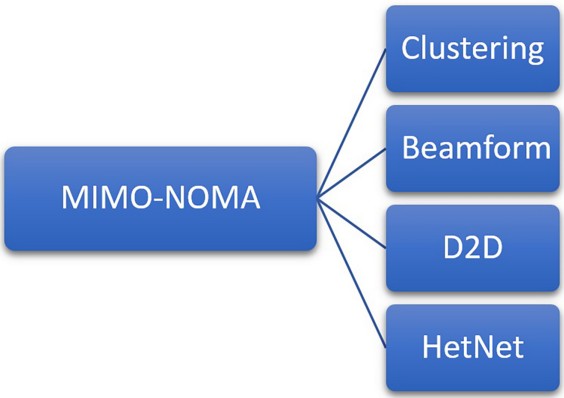

**Figure 6 Techniques and areas for enhancing data rate in NOMA-MIMO configuration.**

directly affected by the number of users in the system. In addition, NOMA-MIMO allowed for other emerging technologies to be used efficiently.

Techniques and areas used for enhancing data rate in NOMA-MIMO configuration are exhibited in Fig. 6. The integration of NOMA-MIMO with clustering, beamforming, D2D, and HetNet presents an opportunity to improve wireless communication systems. D2D communication improves spectral efficiency, coverage, and latency by bypassing the base station and enabling direct connection between adjacent devices, NOMA-MIMO improves spectrum efficiency by allowing several users to use the same time-frequency resources non-orthogonally. Also, clustering is the process of grouping individuals or devices according to certain parameters, such as proximity or channel conditions. And beamforming is used in these clusters to improve communication between users or between cluster members, which will improve system performance. HetNets are made up of a variety of cell types with varying capacities and coverage, including macrocells and small cells. By employing NOMA-MIMO, D2D, beamforming, and clustering techniques, HetNets can improve coverage, reduce interference, maximize network capacity, and perform better overall.

The integration of NOMA-MIMO, D2D communication, beamforming, clustering, and HetNets presents an integrated approach to tackle the challenges of contemporary wireless communication systems, offering increased throughput, improved coverage, enhanced spectral efficiency, and reduced latency.

### Clustering

Clustering divides nodes in a network with similar objectives or behaviors into logical groupings to achieve various network objectives, including enhancing load balancing, social awareness, equitable resource distribution, network longevity, and spectrum efficiency (*Waqas et al., 2020*). Clusters in a 5G network are formed from nodes that control operations at the cluster level, communicate within the cluster, and communicate with neighboring clusters (*Asadi, Wang & Mancuso, 2014*). Figure 7 shows a cluster analogy where logical groups are created and populated by users. By grouping nodes

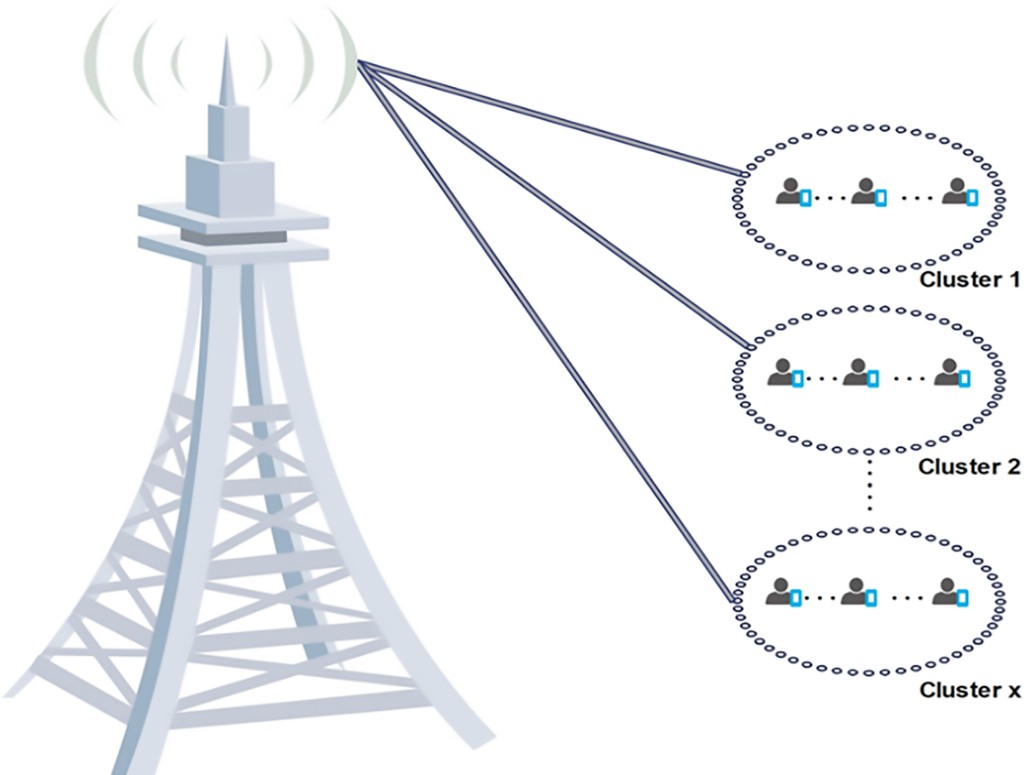

**Figure 7 Cluster analogy with logical groups populated by users.** Image source credit: cell tower and phone icon: Wondershare EdrawMax.

logically according to shared traits or characteristics, clustering enhances network performance, stability, and lifetime (*Hatoum et al., 2011*). It divides heterogeneous network entities into logical groups to reduce interference (*Kumar et al., 2019*; *Gimenez, Michaelsen & Pedersen, 2016*), loosen computation tasks from macrocells (*Joud, Garcia-Lozano & Ruiz, 2018*), and allow for spatial reuse of frequency bands (*Zhu, Zhao & Zhou, 2018*; *Waqas et al., 2020*).

*Saggese, Moretti & Abrardo (2020)* proposed a scheme to minimize power transmitted under the constraint of data rate for each user in a mobile user (MU)-MIMO-NOMA system by optimal clustering, beamforming, and power allocation. Users are grouped in clusters, each containing strong and weak channel users. The problem is considered joint optimization. Under the constraints of user rate, the problem in this work became a fair resource allocation between users. This problem is considered a mixed-integer non-convex problem with no optimal solution. Therefore, the authors divided the problem into three sub-problems:

- Beamforming problem: Block diagonalization (BD) beamforming is used so that the spatial pre-coder eliminates interference between clusters.
- Power allocation problem: This problem is convex, and the Lagrangian dual domain is used to solve it.

- Clustering problem: This is a mixed-integer linear programming (MILP) problem, specifically weighted bipartite matching (WBM), which provided an optimal solution considered as a clustering strategy that resulted in minimum transmit power.

*Kim et al. (2015)* proposed a user clustering algorithm for NOMA taking into account channel correlation between channel gain and users to maximize the sum-rate in MU-MIMO-NOMA setup. The authors used fractional transmit power control (FTPC) as a power allocation method, where power was divided between the near and far user to achieve the maximum sum rate. The use of FTPC guaranteed the rate of each user and small cancellation error of SIC for the near user. The clustering algorithm used in this work, groups users who are aligned in the same direction, this grouping helps to mitigate and avoid multi-user interference (MUI) from the other beams, also the SINR was improved as a result of grouping two users based on the channel gain. Also, to mitigate and suppress MUI this work created an optimization algorithm for the pre-coding matrix based on the majorization minimization (MM) approach which is a non-convex objective function.

*Ali, Tabassum & Hossain (2016)* have done work on NOMA uplink and downlink where a sum-throughput maximization problem in a cluster optimized under power, rate, and SIC constraints. The authors proposed efficient user clustering and power allocation solutions, a two-step approach is used to solve the problem, firstly grouping users into clusters and optimizing users' power allocations, which made this problem of MINLP type. For the clustering problem, a low-complexity sub-optimal user grouping scheme was introduced, where the decision of allocating users inside a cluster is decided based on channel gain differences among users. As for optimal power allocations, for any cluster size and using Karush-Kuhn-Tucker conditions (KKT) the authors found a derived closed-form solution for optimal power allocations in order to maximize the overall system throughput by individually maximizing the sum throughput of users in a given cluster for both uplink and downlink NOMA. Also, the authors in their work indicated that distributing all high-channel gain users into different clusters enhances the sum throughput of all clusters. This is due to grouping high channel gain user with lower channel gain user which has a direct effect on allowing a higher rate for high channel gain user and also for the lower channel gain user. As such, the arranging of users among other clusters in the system should have the second highest channel gain user and the second lowest channel gain user, until all users are arranged to their prospective cluster.

*Ding & Cai (2017)* introduced a linear optimization approach aimed at minimizing overall power consumption among MUs. They then proceed to jointly optimize beamforming vectors and power allocation coefficients for a MIMO-NOMA cluster. The proposed method yields a closed-form expression for the cluster's beamforming matrix, effectively circumventing the impact of peer effects during mobile user clustering. the proposed NOMA cluster beamforming approach provides an efficient way to calculate the beamforming vector and power allocation coefficients for each MIMO-NOMA cluster. This approach is based on linear optimization using zero-forcing beamforming to minimize the cluster power consumption. then the closed-form approach is then used in

order to reduce the overall power consumption in MU clustering. For the power allocation to MU two scenarios were proposed, a power coefficient set or a single power coefficient. The authors in their work managed to avoid the peer effect, which is the result of recalculating beamforming vectors for all MUs due to a change in one mobile user beamforming vector because all mobile user beamforming vectors are calculated at the same time. By combining the value of the beamforming vector and power coefficient values, the authors successfully transform the initial superposed transmit signal into a novel format, consequently establishing a redefined non-zero-forcing (ZF) beamforming problem. They present two key discoveries aiding in the efficient identification of the MU clustering set with minimal total power consumption. MU clustering stands out as a viable strategy for power consumption reduction. Another is that a cluster's maximum power reduction is tied to the MUs in that cluster and unrelated to MUs outside of it.

*Ali, Hossain & Kim (2017)* proposed a multi-cluster zero-force beamforming (ZF-BF) technique and a dynamic power allocation solution for the downlink MIMO-NOMA system. A ZF-BF technique which is a low-complexity sub-optimal user clustering scheme that enhances the sum-spectral efficiency in the cell by exploiting the channel gain differences and correlations between users and also reduce inter-cluster interference in NOMA system. To achieve optimal user clustering authors proposed an exhaustive search among all the users in a cell which increase the computational complexity of user clustering for downlink MIMO-NOMA system. The authors used a decoding scaling weight factor to mitigate inter-cluster interference while simultaneously amplifying the desired signal. This involved transmitting the decoding scaling weight factor to designated users before commencing data streams. In pursuit of maximizing sum-throughput *via* clustering, the authors designated the users with the highest channel gains in a cell as cluster heads. They also selected users with more correlated channel gains to the cluster head to effectively suppress inter-cluster interference and enhance capacity gain. In a multi-cell MIMO-NOMA system, *Ding & Cai (2020)* proposed a two-side coalitional matching approach to optimize resource allocation, based on zero-forcing beamforming authors derive a low-complexity closed-form solution for resource allocation. The authors considered a combined clustering problem the first is BS selection and the second is MIMO-NOMA clustering. The model authors developed consists of several base stations positioned in the cell center, for each antenna element the received signal from each user is written in the form of the weighted sum of the beamforming vectors and transmit power, where different power coefficients are calculated, then SIC decoding is performed and the SINR for each user is obtained. With the objective of maximizing the overall data rate, the authors derived a resource allocation strategy for a MIMO-NOMA cluster in a single BS to maximize the total data rate. As for the MIMO-NOMA clustering problem, authors proposed a three steps coalition game, where clustering user is performed then a split-and-merge rule to decide if a coalition between the user and suitable cluster should be formed, the process is repeated until no user changes its strategy based on normalized channel gain. Authors mentioned in their work other clustering strategies, the correlation-based approach and the gain-difference approach, the two-stage approach proposed outperforms the other two approaches in terms of maximizing the sum data rate. *Liu et al. (2018)*

proposed a cluster-based MIMO-NOMA where clusters have users grouped, and all users within the same cluster share the same beamform. It is noted that cluster-based MIMO-NOMA shows high sum rates where the enhanced MIMO-NOMA scheme was found to perform better. *Khan et al. (2020)* explored the power allocation and user admission optimization problem to optimize the throughput in a multi-cluster MIMO-NOMA downlink system under user QoS constraints.

*Al-Hussaibi & Ali (2019)* proposed a new approach with receive antenna surface (RAS) aimed to maximize the connectivity and sum rate capacity in a massive MIMO-NOMA uplink channel by reducing the numbers of radio frequency chains (RFC) and computational complexity. A joint exhaustive search algorithm is proposed for sum rate maximization and three sub-optimal algorithms are proposed to achieve dynamic user clustering, RAS, and power allocation. With perfect channel state information (CSI), authors clustered users based on the average received signal power, and users are sorted using the calculated channel path loss and placed by multi-user detection mechanism (MUD) in either a high power cluster (HPC) or lower power cluster (LPC), this dynamic user clustering is updated whenever users' locations changed. To maintain user rates during transmission time authors used NOMA-PD to keep average received powers for HPC and LPC. The disparities of user signal due to signal attenuations are compensated using statistics-aware intra-cluster power allocation. *Zhang et al. (2021)* aimed to reduce the interference among users within a MIMO-NOMA cluster by tackling a nonlinear precoding scheme and a clustering scheme, where the weighted coefficients of a user channel information in a given cluster are pre-coded in the transmission power constraint. An alternating direction method of multipliers (ADMM) algorithm is used for the optimization problem. Also, the authors proposed a clustering scheme that will reduce the interference and provide a better bit error rate (BER). The optimization problem is solved to get preceded signals for each user cluster. The use of a linear precoding scheme allows for lessening system complexity, and the ADMM algorithm used is suitable for distributed convex optimization problems. To achieve the expected receiving performance in the MIMO-NOMA downlink authors used squared errors between the superimposed signals and the corresponding receiving signals, and the difference of correlation between the user channel and channel gain is considered thoroughly. Also, to perform SIC decoding efficiently authors depended on the difference between channel correlation and channel gain among users within a cluster. For the power allocation authors considered more the decay factor is larger, the more power is allocated to the user with lower channel gain.

In a MIMO-NOMA multi-cluster system *Bakulin et al. (2023)* compared open-loop and closed-loop systems to compare noise immunity with different types of modulation. In a MIMO-NOMA system with an open loop CSI is not used when forming a precoding matrix which makes power distribution among users ineffective, while in a closed loop, CSI is considered in forming a precoding matrix and thus a power allocation scheme is needed to improve SNR between users and increase the performance. *Zeng et al. (2017b)* aimed to improve the uplink sum rate in a MIMO-NOMA system based on group-level successive interference cancellation (GLSIC) that uses NOMA and space-division multiple access (SDMA) to reduce inter-group interference. The grouping decision is based on user

**Table 3 Summary of works on clustering.**

| Ref. | Objective | Algorithm and type of the problem | | Constraints | Outcomes |
|---|---|---|---|---|---|
| Gimenez, Michaelsen & Pedersen (2016) | Minimize power transmitted | MILP and the problem of type WBM | Mixed-integer non-convex | The data rate for each user | Improved power. Superiority of the clustering algorithm |
| Rihan, Huang & Zhang (2018) | Clustering algorithm to maximize the sum-rate | MM approach for nonconvex objective function | Non-convex objective function | Channel correlation between channel gain and users. Optimizing algorithm considering fairness. | Clustering algorithm improves sum rate and SINR |
| Asadi, Wang & Mancuso (2014) | Sum-throughput maximization | MINLP type. Derived closed-form optimal power allocation. | Low-complexity sub-optimal solution | Power, rate, and SIC | Improves sum-rate and SINR |
| Zhu, Zhao & Zhou (2018) | Maximize connectivity, sum rate in a massive MIMO-NOMA uplink channel | Exhaustive search algorithm | Joint low complexity problem | Received power. Minimum rate requirements. Allowed users. | Increase users. Double utilized RFC. Higher sum rate |
| Joud, Garcia-Lozano & Ruiz (2018) | Reduce the interference among users in a cluster | Linear precoding scheme | Optimization objective | Transmission power constraint. | Better BER. Less complex algorithm |
| Zeng et al. (2017b) | New clustering schema based on GLSIC | MMSE linear detector | | Number of users. Power allocation | Higher uplink rate up to 32 users. |

Note:
MILP, mixed-integer linear programming; WBM, weighted bipartite matching; MM, majorization minimization; SINR, signal-to-interference plus noise ratio; MINLP, mixed integer non-linear programming; SIC, successive interference cancellation; MIMO-NOMA, multiple-input multiple-output non-orthogonal multiple access; RFC, radio frequency chains; BER, bit error rate; GLSIC, group-level successive interference cancellation; MMSE, minimum mean square error.

distances from the base station. The authors compared the proposed GLSIC to the conventional cluster-based strategy which assumes users are clustered based on a spatial direction and different channel gains thus giving a degraded throughput for the system. In their work, authors considered users randomly distributed within the cell and according to propagation distance to the base station users are divided into groups, this allows for transmitting a parallel signal to users within the same group. This schema allows to minimize training overhead when performing channel estimation, where users in the same group use mutually orthogonal pilot. For uplink data transmission at the base station authors used a maximal-ratio combining (MRC) linear detector process, then followed by the GLSIC step to remove inter-group interference where decoding user signal follows an ascending order based on the group indices. Table 3 shows a summary of work done on clustering.

In conclusion, the studies suggest a comprehensive approach to optimizing MIMO-NOMA systems, focusing on clustering, beamforming, power allocation, and interference mitigation for superior performance in various deployment scenarios. *Ali, Tabassum & Hossain (2016)*, *Ali, Hossain & Kim (2017)*, *Saggese, Moretti & Abrardo (2020)*, *Kim et al. (2015)*, *Ding & Cai (2017)*, and *Ding & Cai (2020)* explore clustering, beamforming, and power allocation strategies to minimize power consumption and optimize signal transmission. They also propose innovative algorithms, linear optimization, and multi-

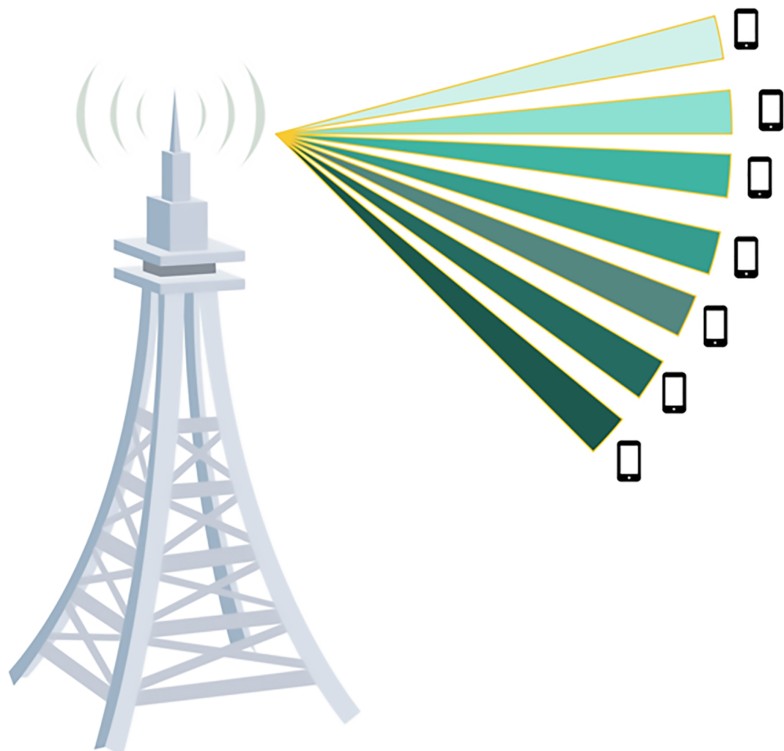

**Figure 8 Beamform MIMO-NOMA system.** Image source credit: cell tower and phone icon: Wondershare EdrawMax.

cluster solutions. Also, *Liu et al. (2018)*, *Al-Hussaibi & Ali (2019)*, *Zhang et al. (2021)*, *Bakulin et al. (2023)*, *He et al. (2023)*, and *Roy et al. (2021)* highlight the importance of refined power allocation, user admission policies, and innovative interference cancellation techniques in improving SINR and system performance. Also, the studies highlight several advanced techniques for optimizing MIMO-NOMA systems. Key strategies include BD beamforming, fractional transmit power control (FTPC), and KKT conditions for reducing interference and ensuring fair power distribution. ZF-BF and RAS improve efficiency and reduce complexity. Two-side coalitional matching optimizes resource allocation by considering both BS selection and clustering. The ADMM algorithm refines power distribution, while GLSIC and SDMA reduce inter-group interference and enhance uplink sum rate. These techniques collectively enhance MIMO-NOMA system performance through improved clustering, beamforming, power allocation, and interference mitigation.

## Beamforming

Beamforming techniques focus on the transmit power of antennas in specific directions, enhancing signal quality and reducing interference. Due to the improvements resulting from implementing MIMO in NOMA systems, the literature introduces two types of beamforming techniques, beamformed MIMO-NOMA and cluster-based MIMO-NOMA. In beamform-based MIMO-NOMA, all user's messages are interfered with in the same network, this schema leads to exponential complexity due to the need to jointly optimize the decoding order and the beamform, taking into account users' numbers (*Zhu et al.,*

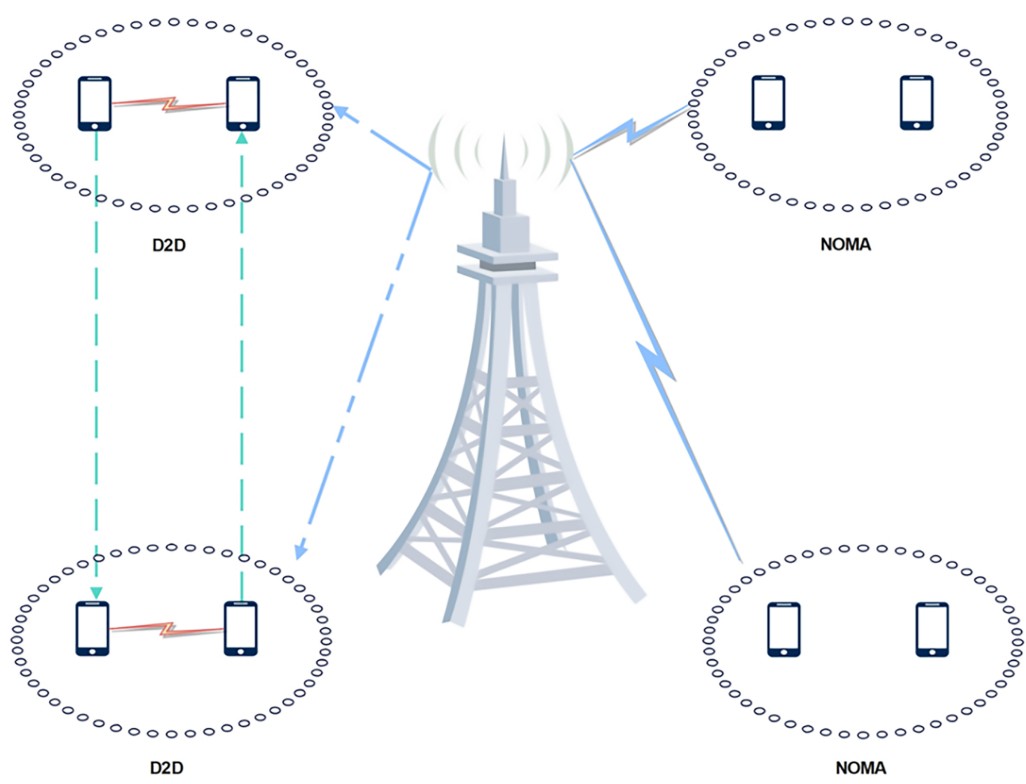

**Figure 9 D2D communication system in a NOMA MIMO configuration.** Image source credit: cell tower and phone icon: Wondershare EdrawMax. 

*2019*; *Li et al., 2022*; *Norouzi, Morsali & Champagne, 2019*; *Lim et al., 2024*). Figure 8 shows a beamform-based MIMO-NOMA system.

Beamforming is a simple transmitting and receiving beamforming technique. Each user has its own linear beamform generated specifically while maintaining constraints that are sufficient to ensure a successful SIC, as shown in Fig. 9. In order to perform a successful, SIC at the user end, the base station needs to ensure robust received power not only for the intended recipient but also for other users scheduled to decode the message before their own in the SIC process. *Sun et al. (2024)* found that beamform-based MIMO is able to perform in overloaded and under-loaded regimes. Similar to SDMA which has paved to road for beamform-based MIMO. Using CSI, *Sun et al. (2015)* maximized the ergodic capacity by transmitting a covariance matrix while maintaining a two-user downlink MIMO-NOMA system in a given decoding order. *Hanif et al. (2016)* proposed a minorization-maximization algorithm for the beamform design after taking into account a multi-user downlink multiple-input single-output (MISO)-NOMA system with a predetermined decoding order, the result obtained in this work suggests that a MIMO-NOMA system that has a number of users larger than the number of transmitting antennas can outperform the traditional ZF beamforming method. *Choi (2016)* proposed a MIMO-NOMA method with layered transmission and tackled the power allocation optimization for sum-rate maximization.

**Table 4  A summary of methods used in MIMO-NOMA beamforming.**

| Ref | Method | Description |
|---|---|---|
| *Choi (2016)* | Linear Beamforming with SDMA | Maintains constraints for successful SIC. Generates unique beamforms. |
|  |  | Operates in overloaded and under-loaded regimes. |
| *Baidas et al. (2019)* | Covariance Matrix Transmission | Two-user downlink MIMO-NOMA system with predetermined decoding order. |
| *Chang et al. (2017)* | Minorization-Maximization Algorithm | Outperforms traditional zero-forcing beamforming. |
|  |  | Utilizes MIMO-NOMA method with layered transmission. |
| *Dai et al. (2019)* | Layered Transmission | Optimizes power allocation for sum-rate maximization. |
| *Waqas et al. (2020)* | User Clustering for SIC Optimization | Allocating users with distinct channel gains to different clusters. |
| *Saggese, Moretti & Abrardo (2020)* | Beamform Optimization under Outage Probability Constraints | Addressing beamform optimization problem under outage probability constraints. |
|  |  | Utilizing successive convex approximation and semi-definite programming. |

Various beamform designs that accept a certain amount of inter-cluster interference have been taken into consideration in several publications. *Ali, Hossain & Kim (2017)* suggested a beamforming method for the MIMO-NOMA downlink that may effectively cancel a significant amount of the inter-cluster interference when the number of antennas transmitting from the base station is smaller than the antennas at the user's side. In addition, an effective user clustering method was proposed in *Ali, Hossain & Kim (2017)* with the aim of maximizing SIC performance by allocating users who have very distinct channel gains to a different cluster. *Cui, Ding & Fan (2018)* addressed the optimization problem of beamforming under outage probability constraints, considering two distinct types of imperfect CSI knowledge. They applied efficient algorithms utilizing successive convex approximation (SCA) and semi-definite programming (SDP) techniques. Table 4 shows a summary of methods used in MIMO-NOMA beamforming.

In conclusion, the studies present various approaches to optimizing MIMO-NOMA systems. Techniques such as BD beamforming, FTPC, and KKT conditions are utilized to reduce inter-cluster interference, ensure fair power distribution, and provide robust solutions for non-linear optimization problems. ZF-BF and RAS enhance efficiency by eliminating inter-cluster interference and reducing RFC and computational complexity. The two-side coalitional matching approach ensures efficient resource allocation by considering both BS selection and MIMO-NOMA clustering. The ADMM algorithm optimizes power distribution, reducing system complexity. GLSIC combined with SDMA effectively reduces inter-group interference, enhancing the uplink sum rate. These concepts and reasoning provide a comprehensive approach to enhancing the performance and efficiency of MIMO-NOMA systems through advanced clustering, beamforming, power allocation, and interference mitigation techniques.

## D2D

Device-to-device (D2D) communications, allow devices to communicate directly with one another in an effort to increase throughput, power consumption, and spectrum efficiency. Another crucial technology being studied for 5G is NOMA, which can improve performance metrics and address resource scarcity issues (*Chang et al., 2017*; *Baidas et al., 2019*; *Dai et al., 2019*; *Benjebbour et al., 2013*; *Palattella et al., 2016*; *Yoon et al., 2018*). Figure 9 exhibits a D2D communication system in a NOMA MIMO configuration.

*Khan et al. (2023)* introduced co-channel interference which is a primary limitation of D2D communications, D2D technology allows users located near each other to communicate directly with one another and bypass BS. Reconfigurable intelligent surfaces (RIS) are a potential technology that improves wireless communications by overcoming interference and manipulating electromagnetic waves in their surroundings. In order to maximize the sum rate of NOMA D2D communications Authors improved RIS D2D communications by optimizing uncrewed aerial vehicle (UAV) and NOMA power budget and passive beamforming of RIS while maintaining quality of service. UAVs can be used in situations of emergencies or the need for high-capacity communication. D2D communications with NOMA and UAVs have demonstrated a lot of promise for improving wireless networks. The optimization problem is divided into two parts due to non-convexity and the authors used alternating optimization to solve the problem. The convexity of the problem comes from the interference in the rate expressions which was reduced using the SCA method; authors then used the Lagrangian method to solve the problem along with the KKT constraints and conditions, also the transmit power at D2D transmitters (DT) and passive beamforming at RIS. Authors used SINRs to reformulate the passive beamforming at RIS and this made the problem convex and used Schur complement to design the matrix of the passive beamforming. *Chrysologou, Chatzidiamantis & Karagiannidis (2022)* proposed a hybrid cellular and bidirectional D2D communication network, BD2D-CNOMA, with two far users assisted by a relaying node where both cellular and bidirectional D2D communications in the same frequency. System performance is evaluated under both perfect and imperfect successive interference cancellation, providing closed-form expressions for outage probabilities and ergodic capacities of all data streams. The proposed scheme aims to enhance network capacity and spectral efficiency while enabling various applications through bidirectional D2D communication. To assist in the transmissions from the users to the BS, a relaying node that follows the decode-and-forward (DF) approach is employed. The transmission protocol consists of three phases, each occupying a single time slot. During each phase, the users send composite signals with power allocation coefficients. The received SINRs at the users and relay are determined by considering the power allocation and decoding principles specific to NOMA. *Pan et al. (2018)* aimed to maximize the sum rate of D2D communication while maintaining the minimum rate requirements for the NOMA system. The authors investigated power control and channel assignment problems with the constraint of keeping the SIC decoding order of the NOMA system. To achieve maximizing the sum rate, authors defined optimal conditions for power control of NOMA

users in the system then an iterative method was proposed to solve the resource allocation problem, the problem became convex by the use of auxiliary variables and relaxing the binary constraints. In their work authors calculated SINR for all users in the system in an effort to maintain the constraint that will preserve SIC decoding order. *Khokar, Bajpai & Gupta (2020)* proposed a selection scheme for a cooperative device-to-device (C-D2D) communication system, where the D2D source that has the highest SINR which is capable of improving cellular transmission is choosing to reuse the spectrum to other devices. To achieve this, authors considered a full-duplex (FD) D2D device scenario that uses the self-interference suppression (SIS) technique to cancel self-interference (SI) which is common in FD systems. Also, the authors used NOMA as a power splitting protocol to maintain equal allocation of resources in cell uplink and D2D transmission. And used successive interference cancellation SIC at the D2D receiver to decode the signals. The authors' scheme improves the performance by choosing the optimal power splitting factor and SIS parameters.

*Solaiman, Nassef & Fadel (2021)* proposed a novel approach for resource allocation to manage and mitigate different types of interference in an intelligent manner. the resource allocation problem is into three sub-problems: interference-aware graph-based user clustering, MIMO-NOMA hybrid beamforming (HB) design, and optimized power allocation utilizing particle swarm optimization (PSO). Authors considered a small-cell downlink MIMO-NOMA to model a D2D communication system along with full knowledge of CSI and minimum power limit to satisfy user quality of service and D2D pairs. For the clustering problem authors propose an algorithm for user clustering based on graph theory in order to enhance the performance of the system. This algorithm aims to reduce the SIC process and identify the optimal user clusters for MIMO-NOMA beamforming. The graph-based user clustering algorithm utilizes graph theory to divide the graph into separate subsets of interconnected vertices, with each vertex representing a user and each edge representing the channel correlation between two users. The weights assigned to the edges reflect the strength of the channel correlation. The algorithm consists of three main phases: user clustering, D2D pair clustering, and cluster matching. By applying this algorithm, the users are grouped into clusters based on their high channel correlations, which facilitates improved beamforming. This clustering approach reduces interference and provides an efficient solution for enhancing the performance of the system. *Li et al. (2020)* introduced a beam space MIMO-NOMA system for broadcasting *via* full-duplex D2D communication. The system considers two users, a near user, and a far user, and communicates using layered division multiplexing (LDM), LDM is a power-based layered division multiplexing that leverages the achievable spectrum efficiency and coverage. In order to increase spectrum efficiency, a two-layer LDM structure was developed, with the lower layer (LL) providing unicast services and the upper layer providing dependable mobile broadcast services. High power is used to transmit the upper layer signal, whereas low power is used to transmit the LL signal. To model channel propagation authors used the Friis equation to represent large-scale fading, and Nakagami-m fading is used to model small-scale fading. *Yu et al. (2021)* aimed to maximize link sum rate, throughput, and capacity through a power optimization strategy designed for D2D

 

communication with imperfect SIC decoding. They formulate a scheme aimed at maximizing the sum data rate while considering the quality of service requirements for cellular users and transmit power constraints. They employed the sub-gradient method to resolve the dual variables, enabling an assessment of the effectiveness of the proposed optimal power allocation approach. Dual theory was utilized to derive an efficient solution to the problem, followed by the application of KKT conditions to address the maximization problem. *Alemaishat et al. (2019)* propose a joint sub-channel and power allocation approach based on NOMA for D2D communication that maximizes the throughput and overall uplink efficiency. graph theory is used to construct weighted bipartite graphs, and the Kuhn-Munkres (KM) algorithm is used to guarantee the quality of cellular users. A channel of the corresponding cellular user is allocated to each D2D group. The authors used Karush-Kuhn-Tucker conditions to obtain the optimal power allocation scheme for D2D users is that each D2D group contains two receiving ends, one strong and one weak, and channels are allocated using NOMA. In their work authors calculated the SINR of the cellular user and the total capacity of the D2D group multiplexing of the channel of cellular users. The total power consumed for each D2D group is calculated based on the ratio of the total capacity of each group. The authors decomposed the optimization problem into two steps, the first is to solve channel allocation and the second is to solve power allocation. The channel allocation problem is converted to the maximum matching problem of the weighted bipartite graph, solved by the KM algorithm. As for the power allocation problem, since each D2D group has an assigned channel, the weak user can reach the maximum value of the limited condition once the strong user's SINR in the group reaches the threshold. Every D2D group's power allocation decision is compared, and the larger one is used as the final allocation scheme. *Cai et al. (2021)* introduced a cooperative relaying system (DRC-NOMA) that uses D2D communications assistance and NOMA to improve spectrum efficiency. The authors introduce a system model for downlink DRC-NOMA and then present a theoretical derivation of the ergodic sum-rate and an optimal power allocation strategy. In their model, a single cell consists of BS, a cellular user, a relay, and a pair of D2D users. BS reaches one user and other users are reached through a rely. The authors used an iterative method and the convergence of the iterative power allocation is presented. A low-complexity power allocation strategy is introduced. Table 5 shows a summary of work done on D2D.

The studies discussed focus on enhancing D2D communications within wireless networks through various optimization strategies. Initially, *Khan et al. (2023)* explored the use of RIS to expand D2D communications, particularly by optimizing UAV and NOMA-PD alongside passive beamforming of RIS, all aimed at maximizing the sum rate and mitigating interference challenges. Furthermore, the work of *Pan et al. (2018)*, *Khokar, Bajpai & Gupta (2020)*, and *Solaiman, Nassef & Fadel (2021)*, dives into the aspects of D2D communication optimization, power control, resource allocation, and interference management strategies. These methods target the maximization of sum rates while ensuring minimum rate requirements are met. Also, *Li et al. (2020)* introduced a beam space MIMO-NOMA system, leveraging layered division multiplexing to enhance spectral

**Table 5 Summary of works on D2D.**

| Ref. | Objective | Algorithm and type of the problem | | Constraints | Outcomes |
|---|---|---|---|---|---|
| *Palattella et al. (2016)* | Maximize the sum rate of NOMA D2D | Optimization problem | Non-convex | Co-channel interference | Optimized power budget |
| *Yoon et al. (2018)* | Enhance network capacity and SE | A DF | | D2D applications SIC constraints | Higher capacity and SE |
| *Khan et al. (2023)* | Maximize sum rate | Logarithmic function | Convex | SIC decoding order | Higher data rate |
| *Chrysologou, Chatzidiamantis & Karagiannidis (2022)* | Selection scheme for C-D2D | Outage probability Rate equations | Resource allocation | SI | Improves performance |
| *Pan et al. (2018)* | Optimized power allocation | PSO | Non-convex MINLP problem | Minimum SINR Power limits | Higher SE and EE |
| *Solaiman, Nassef & Fadel (2021)* | Data rate maximization | Sub-gradient method to | Non-convex optimization problem | QoS Power allocation | Higher data rate |
| *Li et al. (2020)* | Maximize uplink throughput. | Optimization problem | Joint sub-channel and PA algorithm | QoS Power constraint. | Higher UL throughput |

**Note:**
NOMA, non-orthogonal multiple access; D2D, device-to-device; SE, spectral efficiency; DF, decode-and-forward; SIC, successive interference cancellation; C-D2D, cooperative device-to-device; SI, self-interference; PSO, particle swarm optimization; MINLP, mixed-integer non-linear programming; SINR, signal-to-interference-plus-noise ratio; EE, energy efficiency; QoS, quality of service; PA, power allocation; UL, uplink.

efficiency (SE) and coverage, while *Yu et al. (2021)* and *Alemaishat et al. (2019)* proposed power optimization schemes based on sub-channel and power allocation methodologies, respectively, to boost throughput and overall uplink efficiency. Finally, these collective efforts underscore the diverse approaches and complexities involved in optimizing D2D communications within wireless networks to meet the evolving demands of modern communication systems.

## Heterogeneous

Heterogeneous networks and NOMA have emerged as promising network architectures for increasing spectrum capacity and supporting more users. While HetNets holds great promise for 5G, there are certain drawbacks, including cross-tier interference (*Kordi et al., 2020*; *Roslee, Subari & Shahdan, 2011*; *Celik et al., 2019*; *Ghosh et al., 2012*). Sub-frequency bands are used by NOMA-enabled HetNets to increase SE; however, they must strike a balance between increased data rate and power consumption (*Ahmad & Bahadar, 2019*; *Roy et al., 2022*; *Nasser et al., 2021*). NOMA and HetNet have been incorporated to obtain better capacity and coverage, HetNet allows for lesser usage of battery power in user devices. Several NOMA variations have been proposed. Since the main cause of interference in the UL situation is the cell-edge users, resolving the power allocation issue is also necessary to enhance NOMA performance (*Shwetha & Anuradha, 2022*; *Sudhamani et al., 2023*; *Yu & Hou, 2022*; *Lei et al., 2021*; *Hussein, Rosenberg & Mitran, 2022*; *Ullah et al., 2023*; *Rehman, Roslee & Jiat, 2023*). In *Ahmad & Bahadar (2019)* a novel user clustering and power control strategy for NOMA transmission was influenced by
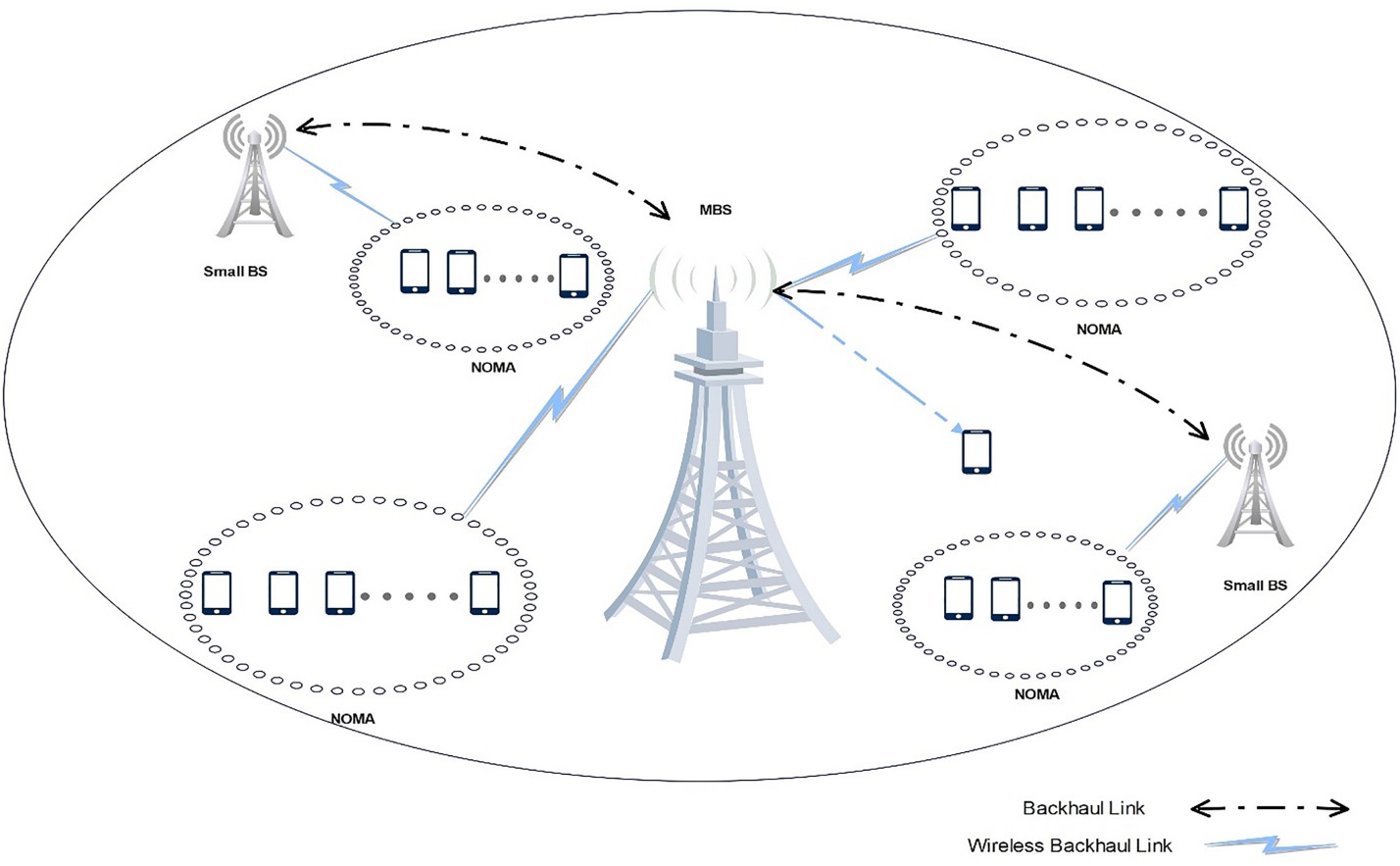

**Figure 10 Heterogeneous network architecture achieving a higher sum rate and better outage performance with lower power back-off value.**
Image source credit: cell tower and phone icon: Wondershare EdrawMax.

HetNet. Figure 10 exhibits a heterogeneous network architecture. This strategy produced a greater sum rate with a lower power back-off value, but it also enhanced outage performance.

*Trankatwar & Wali (2022)* proposed a power allocation scheme for downlink transmission NOMA-based HetNet to optimize the sum rate and outage probability for the system with the constraint of preserving minimal user rate. Using a power allocation coefficient where users' order is based on their channel gain condition, the authors introduced a fast and minimal complexity algorithm that can achieve maximization of the sum rate and minimization of outage probability, the proposed algorithm has a low time complexity where the inner loops and the outer loops are giving executing time that makes the algorithm suitable for real-world wireless communication network.

In addition, *Hadi & Ghazizadeh (2020)* proposed a scheme for a sub-optimal iterative approach to solving sub-channel assignment problems and power allocation for a heterogeneous network based on OMA NOMA schemes. Authors used OMA and NOMA sub-channels to serve small-cell base station (SBS) users and used OMA sub-channels for macrocell base station users, this will help to mitigate cross-tier interference between

macrocell and small-cell and will improve system performance and total sum-rate. The authors' problem was divided into two parts, first the sub-channel and second, the power allocation problem, which made the problem the NP-hard optimization type. The sub-channel assignment problem is tackled by solving the linear problem and using the Lagrange dual method to solve the power allocation method. The authors created power allocation and sub-channel assignment matrix respectively, which ultimately made the objective function convex. *Wu, Li & Shi (2022)* aimed to maximize the sum rate of a heterogeneous NOMA system where users are served has different speeds while adhering to QoS constraints. The authors divided the problem into three subproblems, firstly user grouping, secondly user pairing, and thirdly a power allocation scheme, this made the problem of the non-convex type. In their work, authors grouped users in the same base station based on their channel condition and followed by a many-to-one matching algorithm to resolve the problem of matching users and sub-channels and to tackle the problem of power distribution among users' authors introduced a SCA algorithm to solve the problem. *Hasan, Rizvi & Shabbir (2022)* implemented a clustered NOMA-PD approach within a multi-tier ultra-dense HetNet with the aim of increasing network capacity and throughput gains. Their analysis incorporates a hybrid clustered multi-tier network structure, comprising a solitary macrocell base station alongside clustered SBSs distributed according to a Poisson distribution. To mitigate system interference, a hybrid clustering technique is proposed for the SBSs. Depending on a threshold interference criterion, clusters are categorized as either high-power small-cell base stations (HSBSs) or low-power small-cell base stations (LSBSs). Under power limitations, optimization of power allocation within the clustered HetNet is performed to maximize the sum rate. The process of power adjustment is iteratively conducted from users farther away to those closer, continuing until the furthest user reaches its minimum threshold. The efficiency of densely deploying LSBSs is demonstrated by the incorporation of the received power technique to control user association. The authors tackled cooperative NOMA-PD by considering a scenario where the channel state of the near user is strong. In this case, SIC will be implemented at the near user, and in the second time slot, the near user will act as a relay, forwarding the already decoded information of the far user. This cooperation from the strong user would reduce the throughput of the weak users. This cooperation is necessary since the intervention of the strong user would reduce the throughput of the weak users.

*Wang, Zhang & Jiao (2023)* propose two strategies for NOMA user grouping, the Uniform Block Strategy (UBS) and the Mixed Block Strategy, aimed at distributing resources across different device types based on their performance requirements. They also introduce the Kuhn-Munkres method to achieve optimal grouping in scenarios involving multiple devices. The Mixed Block Strategy facilitates effective coordination among device types with similar bandwidth needs, while the UBS addresses the challenge posed by varying bandwidth requirements.

In the UBS approach, one resource block (RB) is allocated for Types A and B in NOMA transmission. The allocation problem mainly focuses on maximizing network performance by allocating RBs to Type A and Type B devices. In the Type A block, two time-sensitive

devices share radio resources, with the decoding performance significantly influenced by the SIC decoding order. In the Type B block, two throughput-sensitive devices share the same radio resources. In the mixed block regime, two RBs are allotted to two Type A devices and two Type B devices simultaneously under NOMA transmission. The distribution of resources for one RB does not affect the distribution of resources for the other. Using the Monte Carlo approach, authors evaluate three user grouping methods for resource allocation in multi-user scenarios. *Rashid et al. (2023)* aimed to exploit channel diversity for small-cell HetNets with wireless backhaul by utilizing MC-NOMA. The authors developed joint user clustering power and sub-channel allocation problems and used an iterative algorithm called JEEPUS which exploits channel diversity by maintaining the highest possible difference in the transmit power levels in a cluster. Authors considered the ratio of the power allocation as the performance parameter for joint power and sub-channel allocation in uplink MC-NOMA with small-cell HetNets system. Because of the non-linear and fractional structure of the objective function and the non-convex constraints in their work, the optimization problem is non-convex, authors introduced a step-wise approach to reduce complexity. Authors used the SCA technique and the Dinkelbach algorithm to convert the problem's non-convex objective function into a regular parametric form. Users are assigned into sub-channels by the JEEPUS algorithm according to their channel gains and minimum rate needs. *Wang et al. (2020)* introduced a robust power allocation problem tailored for downlink NOMA-based multi-cell HetNets, accounting for incomplete CSI. They devised a worst-case algorithm to transform the objective function and constraints involving uncertain parameters into deterministic ones. Subsequently, they utilized KKT conditions and Lagrange dual methods to address the problem. In their model, the authors consider a downlink two-tier NOMA-based HetNet, where users from macro and small cells overlap. Small-cell users are allowed to access the spectrum utilized by macrocell users as long as the cross-tier interference from each user receiver to the SBS remains within the interference limits of the respective user. To optimize the overall energy of small-cell users, the authors developed a joint user association and power allocation problem.

*Mohajer et al. (2023)* proposed a dynamic optimization model that satisfies the required QoS criteria and maximizes the overall uplink/downlink (UL/DL) to improve throughput. Resource allocation and computational carrier scheduling are separate subproblems. The authors used a mathematical model that has a non-convex nature in order to address the max-min fairness problem, SCA and dual decomposition techniques are used in addition to the sub-gradient method for computing resource allocation. The proposed model also determines the ON/OFF switching frequency of small cells with low computational complexity. The proposed system model is a NOMA MIMO two-tier HetNet with macro and small cells. The authors suggested DC programming and the multi-sided matching technique together with a simple effective carrier matching (CM) algorithm. The algorithm consists of two main processes, the first assigns users to carriers according to their quality where the user with the highest carrier quality is assigned to that particular carrier. and three users who are able to achieve the maximum power index are chosen for the second stage. *Xu et al. (2020)* introduced an industrial cognitive Internet of Things (IoT) system

**Table 6 Summary of works on heterogeneous.**

| Ref. | Objective | Algorithm and type of the problem | | Constraints | Outcomes |
|---|---|---|---|---|---|
| *Sudhamani et al. (2023)* | Optimize sum rate and outage probability | Fast algorithm | A low-complexity | Minimal user rate QoS constraint | Enhancing sum rate and outage probability |
| *Yu & Hou (2022)* | Improve performance and sum-rate | Sub-optimal iterative approach | Convex | Power budget constraint | Improve performance and sum-rate compared to previous configuration |
| *Lei et al. (2021)* | Maximize the sum rate | Three sub-optimal problems | Non-convex type | Quality of service Number of SBS users | Enhanced the balance between computational complexity and performance |
| *Hussein, Rosenberg & Mitran (2022)* | Clustered NOMA-PD scheme | Heuristic algorithm | | Power constraints | Improved connectivity and sum rate |
| *Rehman, Roslee & Jiat (2023)* | Maximize energy efficiency | Dinkelbach algorithm | Non-convex | SIC order | Enhances power consumption |
| | | SCA technique, a step-wise approach | | Minimum data rate | |
| | | | | Minimum QoS | |
| *Trankatwar & Wali (2022)* | Maximization of rate and energy | A worst-case | Non-convex | Cross-tier interference | Optimized rate |
| | | KKT conditions and Lagrange dual methods | | QoS | |
| *Hadi & Ghazizadeh (2020)* | Optimizing UL/DL | DC programming | Non-convex | Fairness criteria | Throughput, power improvement |
| | | Multi-sided matching method | | QoS | |

**Note:**
QoS, quality of service; SBS, small cell base station; NOMA-PD, non-orthogonal multiple access in the power domain; SIC, successive interference cancellation; SCA, successive convex approximation; KKT, Karush Kuhn Tucker; UL/DL, uplink/downlink; DC, difference of convex.

over multi-homing-based cognitive heterogeneous NOMA networks. They proposed a problem involving joint power allocation and secondary IoT device scheduling, considering imperfect CSI and incomplete spectrum sensing. Utilizing successive convex approximation theory and the dual decomposition method, they developed an algorithm for secondary IoT device scheduling and a proportional fairness approach for joint power allocation. *Zhang et al. (2022)* proposed a successful iterative approach for multiuser detection based on SIC. The receiver incorporates the suggested technique and consists of two component detectors. The first detector employs a low-complexity message passing (MP) algorithm for a high-mobility user group with orthogonal time frequency space (OTFS), and the other uses simple frequency-domain equalization (FDE) for a low-mobility user group with orthogonal frequency division multiplexing (OFDM). The outputs of those detectors, which are used in the SIC process, are exchanged iteratively. In the FDE detection, because of its higher power proportion, authors identify the signal of the low-mobility user group in the time frequency (TF) domain and utilize SIC to eliminate interference. By decreasing the noise variance in each cycle, authors apply simple one-tap equalization to recognize the signal of a high-mobility user group symbol by symbol. Table 6 shows a summary of work done on heterogeneous.

The studies from *Trankatwar & Wali (2022)*, *Hadi & Ghazizadeh (2020)*, *Wu, Li & Shi (2022)*, *Hasan, Rizvi & Shabbir (2022)*, *Wang, Zhang & Jiao (2023)*, *Rashid et al. (2023)*, *Wang et al. (2020)*, *Mohajer et al. (2023)*, and *Xu et al. (2020)*, *Zhang et al. (2022)* explore optimization techniques of HetNets with NOMA. *Trankatwar & Wali (2022)* proposed a power allocation scheme in NOMA-based HetNets to optimize sum rate and outage probability while maintaining minimal user rates.

*Lei et al. (2021)* introduced a sub-optimal iterative approach for solving sub-channel assignment and power allocation to mitigate cross-tier interference. *Wu, Li & Shi (2022)* further enhanced sum rates in heterogeneous NOMA by introducing innovative user grouping, pairing, and power allocation strategies. *Hasan, Rizvi & Shabbir (2022)* introduced a clustered NOMA-PD scheme to enhance network capacity in multi-tier HetNets, while *Wang, Zhang & Jiao (2023)* proposed NOMA user grouping strategies for optimized resource allocation in multi-device scenarios using the Kuhn-Munkres method. Additionally, *Rashid et al. (2023)* explored channel diversity in small-cell HetNets through multi-carrier NOMA, introducing iterative algorithms for joint user clustering, power, and sub-channel allocation. Finally, the explored work shows various aspects of power allocation, sub-channel assignment, user grouping, and interference mitigation strategies tailored to NOMA-enabled wireless communication systems. By introducing innovative algorithms and approaches, such as fast and minimal complexity power allocation schemes, sub-optimal iterative methods for sub-channel assignment, and dynamic optimization models for resource allocation and carrier scheduling, these works address the challenges of maximizing sum rates, minimizing outage probabilities, and ensuring QoS constraints in HetNets.

In conclusion, HetNets managing cross-tier interference is critical, and separating NOMA sub-channels for macro and small-cell base stations mitigates this issue. User grouping and pairing based on channel conditions simplify resource allocation and improve performance, while iterative algorithms such as JEEPUS adjust power levels and sub-channel assignments to exploit channel diversity. Robust power allocation techniques address incomplete CSI, ensuring QoS requirements are met. Joint power allocation and device scheduling algorithms optimize resource allocation despite imperfect CSI. These combined approaches significantly improve spectrum efficiency, data rate, and overall network performance in 5G and beyond.

### Resource allocation algorithms

NOMA-MIMO systems rely heavily on sophisticated algorithms for efficient resource allocation, beamforming, clustering, and power control to maximize system performance. These algorithms address various challenges such as interference mitigation, power optimization, user clustering, and dynamic adaptation to changing channel conditions. In NOMA-MIMO setups, optimal clustering schemes ensure fair resource allocation and minimize transmit power by grouping users with diverse channel conditions. Additionally, beamforming techniques like block diagonalization and zero-forcing mitigate interference between clusters and improve signal quality. Power allocation algorithms, including fractional transmit power control and dynamic power allocation, optimize power

**Table 7 Summary of resource allocation algorithms with NOMA MIMO.**

| Ref | Algorithm | Objective |
|---|---|---|
| *Lei et al. (2016)* | Interior Point Method (IPM) | Maximizing NOMA system throughput and sum rate |
| *Salauen, Chen & Coupechoux (2018)* | Penalty Function Method | Maximizing transmitted power in MC-NOMA systems |
| *Kim, Jafarkhani & Lee (2022)*, *Rehman, Roslee & Jiat (2023)*, *Hadi & Ghazizadeh (2020)* | DC Programming | Enhancing NOMA system throughput and system sum rate |
| *Salauen, Chen & Coupechoux (2018)*, *Gupta & Ghosh (2020b)*, *Rehman, Roslee & Jiat (2023)* | Dinkelbach Method | Optimizing power allocation in downlink NOMA cellular systems to maximize system throughput and data rate |
| *Norouzi, Champagne & Cai (2023)*, *Solaiman, Nassef & Fadel (2021)*, *Trankatwar & Wali (2022)* | Gradient Search Algorithm | Maximizing sum-rate and achieving user fairness in partial NOMA systems |
| *Thet & Ozdemir (2020)* | Branch-and-Bound (BB) Algorithm | Joint optimization of user clustering, beamforming, and power allocation to minimize total transmission power while meeting QoS, user clustering, and power constraints |
| *Haq & Taspinar (2021)* | Singular Value Decomposition (GSVD) | Balancing SINR in NOMA-MIMO systems to handle channel estimation errors and channel uncertainty |
| *Rihan, Huang & Zhang (2018)* | Fractional Transmit Power Control (FTPC), Majorization Minimization (MM) | Maximizing sum-rate in MU-MIMO NOMA setup, optimizing pre-coding matrix based on MM approach |
| *Asadi, Wang & Mancuso (2014)*, *Palattella et al. (2016)*, *Yu & Hou (2022)*, *Trankatwar & Wali (2022)* | Karush Kuhn Tucker (KKT) conditions, Gradient Search Algorithm | Maximizing sum-throughput with efficient user clustering and power allocation in NOMA uplink and downlink |
| *Zeng et al. (2017a)* | Linear Optimization, Zero-Forcing Beamforming | Reducing total power consumption in mobile user (MU) clustering, optimizing beamforming vectors and power allocation coefficients |
| *Liu et al. (2018)* | Zero-Force Beamforming (ZF-BF), Exhaustive Search | Maximizing sum-spectral efficiency in downlink MIMO-NOMA system, achieving optimal user clustering |
| *Ali, Hossain & Kim (2017)* | Two-Side Coalitional Matching, Zero-Forcing Beamforming | Optimizing resource allocation in multi-cell MIMO-NOMA system, deriving closed-form solution for resource allocation |
| *Zhu, Zhao & Zhou (2018)* | Joint Exhaustive Search Algorithm, Statistics-Aware Intra-Cluster Power Allocation | Maximizing connectivity and sum rate capacity in massive MIMO-NOMA uplink channel, dynamic user clustering and power allocation |
| *Joud, Garcia-Lozano & Ruiz (2018)* | Alternating Direction Method of Multipliers (ADMM) | Reducing interference and improving BER in MIMO-NOMA cluster, optimizing precoding scheme and clustering |
| *Pan et al. (2018)* | Particle Swarm Optimization (PSO), Graph Theory | Managing interference through interference-aware user clustering, hybrid beamforming, and optimized power allocation |
| *Khokar, Bajpai & Gupta (2020)* | Layered Division Multiplexing (LDM), Friis Equation, Nakagami-m Fading | Improving spectrum efficiency in broadcasting *via* full-duplex D2D communication |
| *Solaiman, Nassef & Fadel (2021)* | Sub-Gradient Method, Dual Theory, KKT Conditions | Maximizing link sum rate, throughput, and capacity in D2D communication with imperfect SIC decoding |
| *Li et al. (2020)* | Weighted Bipartite Graph, Karush-Kuhn-Tucker (KKT), Kuhn-Munkres (KM) | Maximizing throughput and overall uplink efficiency through sub-channel and power allocation in NOMA for D2D communication |
| *Sudhamani et al. (2023)* | Fast Algorithm, Power Allocation Coefficient, KKT Conditions | Optimizing sum rate and outage probability in downlink transmission NOMA-based HetNet with minimal user rate constraint |

(Continued)

| Table 7 (continued) | | |
| --- | --- | --- |
| Ref | Algorithm | Objective |
| *Yu & Hou (2022)* | Lagrange Dual Method, Convex Optimization, OMA, NOMA | Optimizing sub-channel assignment and power allocation for heterogeneous network based on OMA and NOMA |
| *Hussein, Rosenberg & Mitran (2022)* | Monte Carlo Method, Power Shifting, Successive Interference Cancellation (SIC) | Enhancing network capacity and throughput gain in multi-tier ultra-dense HetNet through clustered NOMA-PD |
| *Ullah et al. (2023)* | Uniform Block Strategy (UBS), Mixed Block Strategy (MBS), Kuhn-Munkres Method | Optimizing resource allocation in multi-device scenarios with NOMA user grouping and matching strategies |
| *Rehman, Roslee & Jiat (2023)* | Joint User Clustering, JEEPUS Algorithm, Successive Convex Approximation (SCA) | Exploiting channel diversity in small-cell HetNets with MC-NOMA for enhanced performance and throughput |
| *Trankatwar & Wali (2022)* | Worst-Case Algorithm, Lagrange Dual Method, KKT Conditions | Robust power allocation for downlink NOMA-based multi-cell HetNets with incomplete CSI |

distribution among users to maximize sum rates while maintaining user quality of service. These algorithms often involve solving complex optimization problems, such as mixed-integer non-linear programming and successive convex approximation, to achieve efficient resource allocation. Moreover, techniques, like receive antenna surface and cooperative relaying, enhance connectivity and capacity by reducing system complexity and improving spectrum efficiency. Overall, algorithms play a pivotal role in shaping the performance and efficiency of NOMA-MIMO systems by addressing key challenges and optimizing system resources.

Table 7 shows various algorithms used in MIMO-NOMA systems to optimize connectivity, sum rate capacity, and reduce interference. These algorithms include joint exhaustive search, dynamic user clustering, receive antenna surface optimization, power allocation, nonlinear precoding, clustering schemes, SIC, beamforming, sub-channel allocation, and user grouping strategies.

The significance of the methods and algorithms used in NOMA-MIMO systems can be attributed to several technical reasons:

- Interference mitigation: Techniques such as block diagonalization, zero-forcing beamforming, and receive antenna surface optimization are crucial for minimizing interference in densely packed network environments. By reducing interference, these methods improve signal quality and ensure more reliable communication.
- Power optimization: Algorithms like fractional transmit power control, dynamic power allocation, and the Dinkelbach method are essential for optimizing power distribution among users. Efficient power allocation maximizes the sum rate and ensures that user devices operate within their power limits, which is particularly important for extending the battery life of mobile devices.
- User clustering and grouping: Effective clustering schemes and dynamic user clustering ensure fair resource distribution and reduce transmission power by grouping users

according to their channel conditions. This approach enhances network efficiency and boosts overall system performance.

- Dynamic adaptation: Algorithms such as successive convex approximation and mixed-integer non-linear programming allow for dynamic adaptation to fluctuating channel conditions. This adaptability is crucial in real-world scenarios where channel conditions can change rapidly.

- Beamforming techniques: Advanced beamforming techniques like zero-forcing beamforming and exhaustive search for optimal user clustering enhance spectral efficiency and signal quality. By directing the signal towards intended users and nullifying interference toward others, these techniques improve network performance.

- Resource allocation: Methods such as the IPM, DC programming, and the branch-and-bound algorithm ensure efficient resource allocation by solving complex optimization problems. These methods help in maximizing throughput, sum rate, and spectral efficiency while maintaining QoS constraints.

- Handling imperfect CSI: Techniques such as the Lagrange dual method and robust power allocation algorithms are designed to operate effectively even with incomplete or imperfect CSI. This robustness is critical for maintaining reliable communication and optimizing performance in real-world environments where perfect CSI is rarely available.

- Enhancing connectivity and capacity: Cooperative relaying and receiving antenna surface techniques improve connectivity and capacity by reducing system complexity and enhancing spectrum efficiency. These methods are particularly useful in scenarios like D2D communications and HetNets where network density is high.

- Improving spectrum efficiency: Methods such as LDM and Nakagami-m fading are used to enhance spectrum efficiency, which is essential for supporting a large number of users and devices in modern communication networks.

These algorithms aim to address the challenges of maximizing sum rates, minimizing outage probabilities, and ensuring QoS in NOMA-MIMO systems, also, to enhance the performance of MIMO-NOMA systems in various scenarios, such as uplink/downlink channels, cooperative relaying, D2D communications, and HetNets.

## SINR

SINR is vital for optimizing resource allocation in 5G. Various methods aim to enhance SINR through proper pairing, power allocation, and beamforming. NOMA uses an SIC receiver to decode messages, strong user messages are decoded first and weak user messages are detected as noise. However, incorrect detection can propagate errors. In most of the literature, pairing is based on the distinction of channel gain between users and thus different power levels are allocated. Proper pairing and power allocation (PA) methods are required for efficient resource utilization, reducing interference, and enhancing system capacity. In the literature work (*Al-Basit et al., 2017*; *Wang, Wang & Xiao, 2019*; *Das, Bapat & Das, 2019*; *Xu & Zhang, 2020*) we introduce, explain, and elaborate resource

allocation schemes. *Ding & Cai (2020)* proposed a SDP based optimization approach to maximize users' minimal received SINRs. Authors obtain a robust optimal beamforming solution by using the S-Procedure and a linear matrix inequality (LMI) to maximize a nonconvex optimization issue of a minimum of received SINRs of users. The SINR of a cell is maximized by creating a quasi-convex optimization problem and using bisection search, randomization is utilized to obtain an optimum solution in the event that the beamforming result is not a rank-1. This work tackles the complexity of resource allocation in large-scale multi-cell systems, suggesting a two-side coalitional matching approach to address this issue. Also, discusses resource allocation for a single cluster but does not address scalability for larger networks with multiple clusters, potentially introducing additional complexities. the concept of "relative fairness" as a new objective function, aiming to balance individual MU benefits and network performance. However, its performance compared to other optimization criteria is unclear, and the trade-offs and limitations are not thoroughly discussed.

Furthermore, *Xu & Zhang (2020)* focused on maximizing the minimum achievable SINR across all users within a single-cell downlink millimeter-wave (mmWave) large NOMA system using random directional beamforming (RDB). In this setup, users are paired based on their azimuth angles, and each pair is served by a beam with matching channel vectors and proximity in the beam direction. RDB aids in reducing interference and enhancing beamforming gain, consequently improving SINR for users. The authors introduced the sum of power allocation coefficients based iterative algorithm (SPACIA) to determine the optimal solution and derived closed-form expressions for power allocation coefficients about maximizing the minimum SINR. In their model, the power allocation vector for a cluster equals the sum of signals for that cluster, and the weighted sum of cluster signals is transmitted to all users in the cell. To maximize the minimum achievable SINR, authors utilize an analog precoding vector and assign equal power to each cluster, formulating a convex problem. This problem is decomposed into two sub-problems: optimizing power allocation coefficients and determining the optimal coefficient for maximizing minimum SINR using the SPACIA. The proposed algorithm faces high complexity due to subproblems like interference user clustering, MIMO-NOMA beamforming design, and power allocation, potentially impacting computational efficiency and scalability in large-scale networks. SINR plays a crucial role in assessing the performance of the system. By optimizing power allocation and interference, the goal is to improve the SINR for D2D communications. The proposed algorithm aims to improve beamforming and reduce interference by clustering users with high channel correlations. This approach enhances SINR, a crucial metric for evaluating the algorithm's effectiveness in enhancing the quality and reliability of D2D communications. *Wang, Wang & Xiao (2019)* suggested a novel power allocation scheme for two-user downlink NOMA systems with imperfectly estimated channels, which maximizes the minimal approximated user capacity when faced with a total power constraint. For a two-user MIMO NOMA system, the authors introduced two iterative power allocation algorithms and a closed-form power allocation solution for a NOMA-based multiuser BF system. The authors stated when SINR balancing is applied, the optimal power allocation factor for a two-user downlink

NOMA system with poor channel estimate can be obtained by solving a quadratic equation. The max-min optimization problem can be reformulated as maximizing the minimum approximated user capacity, which is equivalent to maximizing the minimum approximated received SINR. This work compares a proposed power allocation scheme with a fixed approach and a max-min method under perfect channel state information, but the comparison may not cover all possible schemes or performances. The proposed scheme's performance is evaluated through computer simulations, but the lack of experimental validation using real-world hardware and communication setups hinders practical assessment. *Das, Bapat & Das (2019)* proposed a novel uplink NOMA Random Access (NRA) mechanism where sub-bands are assigned to devices based on their signal being properly decoded by the base station over an observation window. In their work authors divided the resources into two groups, one with better received signals and the other left for other devices to contend for transmission. The authors' method achieved better QoS by isolating each group of sub-bands from one another. Additionally, equipment operating in reserved sub-bands appears to consistently get good radio conditions and does not introduce extra interference into non-reserved sub-bands. The proposed mechanism for sub-band clustering relies on SINR measurements to classify channels, but dynamic changes in channel conditions raise concerns about its accuracy over time. Additionally, continuous monitoring of SINR and sub-band assignments may lead to increased system overhead, potentially impacting efficiency. While the mechanism promises differentiated services, trade-offs between fairness, resource utilization, and overall system performance must be thoroughly evaluated, particularly regarding throughput and latency. Table 8 shows a summary of work done on NOMA-MIMO SINR.

Furthermore, the studies address resource allocation and optimization in wireless communication systems, with a focus on maximizing SINR and ensuring fairness among users. *Ali, Hossain & Kim (2017)* employed semidefinite programming and quasi-convex optimization for large-scale multi-cell systems, but scalability concerns and the performance of the "relative fairness" objective remain unclear. *Das, Bapat & Das (2019)* targeted maximizing minimum achievable SINR in single-cell downlink mmWave NOMA, facing challenges in interference clustering and power allocation complexity. *Wang, Wang & Xiao (2019)* introduced power allocation schemes for two-user downlink NOMA, lacking comprehensive performance comparison and validation. Lastly, a study (*Das, Bapat & Das, 2019*) proposed an uplink NOMA Random Access mechanism for QoS enhancement, highlighting concerns about SINR measurement accuracy and system overhead. These works emphasize the importance of optimizing resource allocation while considering practical implementation challenges.

## Open research problem

In the NOMA-MIMO system, the process of pairing users inside clusters depends on the user's channel condition and thus the SINR of each user, *Lei et al. (2016)* designed a matrix representing users to be paired in a cluster based on their channel state. With NOMA, in order to perform successful decoding, there has to be a good amount of interference, which is the reason why when pairing users in a cluster it has to contain strong and weak channel

**Table 8 Summary of works on NOMA-MIMO SINR.**

| Ref. | Objective | Algorithm and type of the problem | | Constraints | Outcomes |
|---|---|---|---|---|---|
| *Wang, Wang & Xiao (2019)* | Robust optimal beamforming | DSP<br>Bisection | Nonconvex optimization problem | Power constraints | Maximize SINR |
| *Das, Bapat & Das (2019)* | Maximize the minimum SINR | SPACIA algorithm | Convex maximization problem | Power budget<br>Lower SINR threshold | PA with SINR balancing |
| *Xu & Zhang (2020)* | Maximizes the minimum user capacity | Quadratic equation | | SIC conditions constraints | Better BER performance |

users. The clustering mechanism in most of the authors' work (*Ding et al., 2017*; *Huang et al., 2019*; *Hanif & Ding, 2019*; *Gupta & Ghosh, 2020a*; *Hong et al., 2020*; *Zeng et al., 2017b*; *Salaün, Coupechoux & Chen, 2019*; *Saggese, Moretti & Abrardo, 2020*; *Lei et al., 2016*; *Salauen, Chen & Coupechoux, 2018*; *Kim, Jafarkhani & Lee, 2022*; *Gupta & Ghosh, 2020b*; *Norouzi, Champagne & Cai, 2023*; *Thet & Ozdemir, 2020*; *Salaun, Coupechoux & Chen, 2020*; *Haq & Taspinar, 2021*) is based on the channel condition and ignores the SINR value. Equation (6) shows the effect of transmitted power on the SINR term, and the proportional relation of SINR to data rate. The proposed method in *Wang, Wang & Xiao (2019)* focused on achieving SINR balancing between users to ensure fairness and enhance overall performance. It presents a closed-form power allocation formula derived from solving a quadratic equation, facilitating efficient implementation. Comparative analysis with other approaches, including fixed power allocation and methods under perfect channel state information, is conducted to assess performance in terms of SINR and its impact on BER performance. *Wang, Wang & Xiao (2019)* emphasized the fundamental role of SINR in formulating the power allocation problem, deriving solutions, and evaluating performance.

In NOMA-MIMO systems, effectively calculating and utilizing SINR is crucial for optimizing data rates and overall system performance. Key research challenges include integrating SINR into user pairing and clustering mechanisms, which currently focus only on channel conditions. Future research should develop accurate SINR matrices and generate SINR-based preference lists to improve system performance. Optimizing power allocation by adjusting power levels and SINR thresholds is vital for achieving higher data rates and fairness. Research should explore power allocation strategies that consider SINR and investigate how different SINR thresholds and beamforming strategies affect SINR levels experienced by mobile units. Leveraging SINR in clustering, base station selection, and resource allocation can significantly improve system performance and fairness. We found that correct tuning of power and appropriate SINR thresholds can enhance data rates, but the examined literature shows limited use of SINR thresholds in power allocation and clustering algorithms. Clustering and base station selection can leverage SINR levels to ensure adequate QoS, while SINR influences resource allocation decisions, optimizing system performance and fairness. Additionally, beamforming strategies are tailored to

enhance desired signals and mitigate interference, directly impacting SINR levels experienced by mobile units. Addressing these research challenges will optimize NOMA-MIMO systems to meet the high data rate requirements of next-generation networks, ensuring enhanced performance and user experience. Therefore, incorporating SINR into the design of pairing and clustering mechanisms is recommended to enhance data rates in NOMA-MIMO systems.

## CONCLUSION

NOMA has been identified as one of the most important and promising technologies for meeting the demands of 5G. NOMA-PD optimizes service by dynamically allocating power, serving multiple users in the same resource block, and improving SE and data rates, particularly for cell edge users. This survey covers the NOMA concept, advantages, and solutions specifically in the power domain and the integration with emerging technologies. This survey introduces funneled work done on NOMA-MIMO to enhance data rate by optimizing resource allocation. The integration of NOMA with MIMO improves data rate and SE by allowing several users to use the same time-frequency resources non-orthogonally. By employing NOMA-MIMO, we can improve coverage, reduce interference, and maximize network capacity. A joint resource allocation problem is solved by algorithms such as Dinkelbach, Lagrange, and DC programming. Also, clustering, pairing, and channel assignment are solved with linear optimization, zero-forcing beamforming, and exhaustive search. such algorithms and techniques are used to obtain optimized solutions in different deployment scenarios such as MIMO, HetNet, and D2D networks. Furthermore. An open research problem is introduced where key implementation and deployment using SINR can be examined. Finally, we anticipate that this survey will provide guidelines for future work related to enhancing data rate in NOMA MIMO systems.

### Funding
This work is supported and funded by a Telekom Malaysia Research & development (TMR&D) grant, RDTC: 241126, MMUE/240081, TM, Malaysia. The funders had no role in study design, data collection and analysis, decision to publish, or preparation of the manuscript.

### Grant Disclosures
The following grant information was disclosed by the authors:
Telekom Malaysia Research & development (TMR&D), RDTC: 241126, MMUE/240081.

### Competing Interests
Sufian Mousa Mitani is an employee of Telekom Malaysia Research & Development. Osama Abuajwa is an employee of Telekom Malaysia Research & Development. Anwar Osman is an employee of Rohde & Schwarz.

## Author Contributions

- Murad Halabouni conceived and designed the experiments, performed the experiments, analyzed the data, prepared figures and/or tables, authored or reviewed drafts of the article, and approved the final draft.
- Mardeni Roslee analyzed the data, authored or reviewed drafts of the article, and approved the final draft.
- Sufian Mitani analyzed the data, authored or reviewed drafts of the article, and approved the final draft.
- Osama Abuajwa conceived and designed the experiments, analyzed the data, authored or reviewed drafts of the article, and approved the final draft.
- Anwar Osman analyzed the data, authored or reviewed drafts of the article, and approved the final draft.
- Fatimah Zaharah binti Ali analyzed the data, authored or reviewed drafts of the article, and approved the final draft.
- Athar Waseem analyzed the data, authored or reviewed drafts of the article, and approved the final draft.

## Data Availability

This is a literature review.

## Supplemental Information

Supplemental information for this article can be found online at http://dx.doi.org/10.7717/peerj-cs.2388#supplemental-information.

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
