# Peer review of "NOMA-MIMO in 5G network: a detailed survey on enhancing data rate"

_PeerJ Computer Science, doi:10.7717/peerj-cs.2388_

## Round 0.1 · original submission · Major Revisions

Reviewers have raised some serious concerns about the methodology of the literature review, the references included in it and its novelty. Please, address these issues properly, specially, authors must address the aspects that make this review different from the one mentioned by reviewer 3 (published in 2023). Not addressing these aspects properly might lead to a reject in the second review process.

Reviewer 1 ·

Basic reporting

The authors addressed the combination of MIMO-NOMA in a survey, they have mentioned important concepts and equations, however, there still a number of important aspects and references they did not cover very well. In addition, there are typos within the paper, not much, but effective and need to be revised. To the authors, kindly check the following recommendations:
There is a clear issue with the technical depth of important sections and subsections within the paper, for instance, in "3.5. Resource Allocation Algorithms", the explanation is general and there is no technical expalanation of the adopted concepts in details and there should be more justifications provided. Please carefully consider revising the whole manuscript to check this issue.

1. There are few misspelled words, such as "analyse" in the abstract. Revise and correct. In addition, there are few acronyms that have been defined afterwords and not on their first appearence, such as Interference-plus-Noise Ratio (SINR), etc.
2. There is a number of important refernces in literature, that have discussed similar topic and mentioned similar findings, however, the authors didn't cite them. For instance, the authors states the following in section 2.1:
" NOMA system
In NOMA-PD the base station will assign two users on the same channel, which is translated to two users allocated the same frequency at the same time. user 1 and user 4 will be allocated to channel 1 and user 2 and user 3 will be allocated to channel 2. The power for channel 1 will be P_n=P_1,1+ P_1,4 and the power for channel 2 will be P_n=P_2,2+ P_2,3. "

The above concept is called user-pairing in NOMA and it is published in:
Z. Q. Al-Abbasi and D. K. C. So, "User-Pairing Based Non-Orthogonal Multiple Access (NOMA) System," 2016 IEEE 83rd Vehicular Technology Conference (VTC Spring), Nanjing, China, 2016, pp. 1-5, doi: 10.1109/VTCSpring.2016.7504524. keywords: {NOMA;Resource management;Bandwidth;Silicon carbide;IP networks;Complexity theory;Multiplexing},

Z. Q. Al-Abbasi and D. K. C. So, "Resource Allocation in Non-Orthogonal and Hybrid Multiple Access System With Proportional Rate Constraint," in IEEE Transactions on Wireless Communications, vol. 16, no. 10, pp. 6309-6320, Oct. 2017, doi: 10.1109/TWC.2017.2721936.
keywords: {NOMA;Resource management;Optimization;Bandwidth;Multiplexing;Downlink;Throughput;Orthogonal frequency division multiple access (OFDMA);non-orthogonal multiple access (NOMA);hybrid multiple access;vertical pairing;sum rate maximization},


Z. Q. Al-Abbasi and D. K. C. So, "Power allocation for sum rate maximization in non-orthogonal multiple access system," 2015 IEEE 26th Annual International Symposium on Personal, Indoor, and Mobile Radio Communications (PIMRC), Hong Kong, China, 2015, pp. 1649-1653, doi: 10.1109/PIMRC.2015.7343563. keywords: {Resource management;Complexity theory;Mobile computing;Wireless networks;Land mobile radio;Bandwidth;Orthogonal Frequency Division Multiple Access (OFDMA);Non-Orthogonal Multiple Access (NOMA);power allocation;sum rate maximization;coverage probability},

However, the respectful authors have not cite any of those references, kindly recheck if there is a similarity as I mentioned. In addition, the equations from (1) to (7) also very similar to those in the above references!.

Experimental design

Overall, the study is well designed, however, there is a lack of technical depth over several parts within the manuscript. Please discuss the technical concepts of the mentioned algorithms and provide technical reasonings.

Validity of the findings

The paper is a survey which mention the findings of other works. So the provided findings are valid.

Cite this review as

·

Basic reporting

no comment

Experimental design

no comment

Validity of the findings

no comment

Additional comments

The work is generally important and quite sufficient. It may become more important after a few changes:
The manuscript should be written justified.
Multiple access methods should be compared either graphically or in a table.
A table on future work should be given.

Cite this review as

Reviewer 3 ·

Basic reporting

Clear and unambiguous, professional English used throughout: No

Literature references, sufficient field background/context provided: No

Professional article structure, figures, tables. Raw data shared: Yes

Is the review of broad and cross-disciplinary interest and within the scope of the journal? Yes

Has the field been reviewed recently? If so, is there a good reason for this review (different point of view, accessible to a different audience, etc.)? Yes

Does the Introduction adequately introduce the subject and make it clear who the audience is/what the motivation is? No

Formal results should include clear definitions of all terms and theorems, and detailed proofs.

The survey on MIMO-NOMA is already published in 2023 which is quite similar to this. Also, the structure of the paper is not impressive. Figures quality is low not clear text in the figures.

Experimental design

Article content is within the Aims and Scope of the journal: Yes

Rigorous investigation performed to a high technical & ethical standard: No

Methods described with sufficient detail & information to replicate: No

Is the Survey Methodology consistent with a comprehensive, unbiased coverage of the subject? If not, what is missing? Yes But (To ensure unbiased coverage, the methodology could benefit from explicitly incorporating a diverse range of sources, including academic research, industry reports, and real-world case studies, to provide a balanced perspective. It should also involve a systematic review process, possibly with predefined criteria for selecting and evaluating sources, to avoid any potential bias in the selection of literature. Furthermore, including expert interviews or stakeholder surveys could enrich the analysis by incorporating practical insights and diverse viewpoints.)

Are sources adequately cited? Quoted or paraphrased as appropriate? Yes but more recent references are missing.

Is the review organized logically into coherent paragraphs/subsections? Not a comprehensive survey.

Validity of the findings

Impact and novelty not assessed. Meaningful replication encouraged where rationale & benefit to literature is clearly stated: No

Conclusions are well stated, linked to original research question & limited to supporting results: No

Is there a well developed and supported argument that meets the goals set out in the Introduction? No

Does the Conclusion identify unresolved questions / gaps / future directions? No

Cite this review as

---

## Round 0.2 · accepted · Accept

The paper can now be published.

Reviewer 1 ·

Basic reporting

The authors have well responded to my recommendations. I have no further comments.

Experimental design

The design and the whole paper are now suitable for publication.

Validity of the findings

The findings are valid.

Cite this review as

Reviewer 3 ·

Basic reporting

Clear and unambiguous, professional English used throughout. No

Literature references, sufficient field background/context provided: Yes

Professional article structure, figures, tables. Raw data shared: No

Is the review of broad and cross-disciplinary interest and within the scope of the journal? Yes

Has the field been reviewed recently? If so, is there a good reason for this review (different point of view, accessible to a different audience, etc.)? Yes

Does the Introduction adequately introduce the subject and make it clear who the audience is/what the motivation is? No

Formal results should include clear definitions of all terms and theorems, and detailed proofs.

The survey on MIMO-NOMA is already published in 2023 which is quite similar to this. Also, the structure of the paper is not impressive. Figures quality is low not clear text in the figures.

There are many grammatical mistakes and lack of professional handling of the writing.
There exists survey on the same topic in 2024, one reference is given below.
Apiyo, A., & Izydorczyk, J. (2024). A Survey of NOMA-Aided Cell-Free Massive MIMO Systems. Electronics, 13(1), 231.

Overall the structure of the paper is very non-professional.
The authors claimed that the questions in start will be addresses but unfortunately, I am unable to understand the answers of the questions because of very non-professional writing.

Experimental design

Article content is within the Aims and Scope of the journal and article type. Yes
Rigorous investigation performed to a high technical & ethical standard. No

Methods described with sufficient detail & information to replicate. No

Is the Survey Methodology consistent with a comprehensive, unbiased coverage of the subject? If not, what is missing? Yes But (To ensure unbiased coverage, the methodology could benefit from explicitly incorporating a diverse range of sources, including academic research, industry reports, and real-world case studies, to provide a balanced perspective. It should also involve a systematic review process, possibly with predefined criteria for selecting and evaluating sources, to avoid any potential bias in the selection of literature. Furthermore, including expert interviews or stakeholder surveys could enrich the analysis by incorporating practical insights and diverse viewpoints.)

Are sources adequately cited? Quoted or paraphrased as appropriate? No

Is the review organized logically into coherent paragraphs/subsections? Not a comprehensive survey.

Validity of the findings

Impact and novelty not assessed. Meaningful replication encouraged where rationale & benefit to literature is clearly stated: No

Conclusions are well stated, linked to original research question & limited to supporting results: No

Is there a well developed and supported argument that meets the goals set out in the Introduction? No

Does the Conclusion identify unresolved questions / gaps / future directions? No

Cite this review as